# Direct 4D printing of ceramics driven by hydrogel dehydration

Rong Wang [1,2,5], Chao Yuan [3,5] ✉, Jianxiang Cheng[1,2,5], Xiangnan He[1,2], Haitao Ye[1,2,4], Bingcong Jian[1,2], Honggeng Li[1,2], Jiaming Bai[2] & Qi Ge [1,2] ✉

4D printing technology combines 3D printing and stimulus-responsive materials, enabling construction of complex 3D objects efficiently. However, unlike smart soft materials, 4D printing of ceramics is a great challenge due to the extremely weak deformability of ceramics. Here, we report a feasible and efficient manufacturing and design approach to realize direct 4D printing of ceramics. Photocurable ceramic elastomer slurry and hydrogel precursor are developed for the fabrication of hydrogel-ceramic laminates via multimaterial digital light processing 3D printing. Flat patterned laminates evolve into complex 3D structures driven by hydrogel dehydration, and then turn into pure ceramics after sintering. Considering the dehydration-induced deformation and sintering-induced shape retraction, we develop a theoretical model to calculate the curvatures of bent laminate and sintered ceramic part. Then, we build a design flow for direct 4D printing of various complex ceramic objects. This approach opens a new avenue for the development of ceramic 4D printing technology.

4D printing is an advanced additive manufacturing technology that combines 3D printing and stimulus-responsive materials to create structures that can change their shapes over the fourth dimension "time" in response to environmental stimuli, such as heat[1–5], light[6–8], humidity[9,10], water[11–13], chemicals[14,15], magnetic field[16–18], and others[19,20]. Beyond the conventional 4D printing which requires additional programming step, direct 4D printing[21,22] allows printed flat patterns to directly evolve into complex 3D geometries upon environmental stimulus, and is considered as an easy, fast and economical manufacturing strategy that requires less time and no support material to fabricate 3D structures. So far, 4D printing has been successfully applied to fabricate structures made of hydrogels[23,24], shape memory polymers (SMPs)[25,26] and liquid crystalline elastomers (LCEs)[27]. However, reports about 4D printing of ceramics are rare[28–30].

Ceramic materials exhibit excellent mechanical properties as well as high-temperature and corrosion resistance, and therefore have

been broadly applied to various applications such as aerospace[31,32], electrical industry[33], tissue engineering[34], and many other fields[35,36]. Due to the inherent brittleness and high hardness, machining of ceramic components has always been difficult. 4D printing has great potential in efficiently manufacturing ceramic parts and components with complex structures. To date, there are mainly two strategies to realize 4D printing of ceramics. One approach is to utilize anisotropic shrinkage in sintering process to generate shape change of printed ceramic parts. In this approach, sequential 3D printing of ceramic resins with different solid contents was applied to generate anisotropic shrinkage and consequent shape change during the sintering process[28,29,37]. Reshaping 3D printed green bodies with the assistance of external forces or molds combined with a subsequent sintering process is another effective approach for 4D printing of ceramics. 4D printing of elastomer-derived ceramics (EDCs) has been realized through direct ink writing (DIW) followed by subsequent

¹Shenzhen Key Laboratory of Soft Mechanics & Smart Manufacturing, Southern University of Science and Technology, Shenzhen 518055, China. ²Department of Mechanical and Energy Engineering, Southern University of Science and Technology, Shenzhen 518055, China. ³State Key Laboratory for Strength and Vibration of Mechanical Structures, Department of Engineering Mechanics, Xi'an Jiaotong University, Xi'an 710049, China. ⁴Department of Mechanical Engineering, City University of Hong Kong, Kowloon, Hong Kong SAR, China. ⁵These authors contributed equally: Rong Wang, Chao Yuan, Jianxiang Cheng. ✉ e-mail: chao_yuan@xjtu.edu.cn; geq@sustech.edu.cn

programming steps[30]. Programmable self-shaping could be implemented through printing designed patterns with $ZrO_2$ nanoparticle reinforced poly(dimethylsiloxane) (PDMS) on the prestretched elastomer substrate, and subsequent release of prestretch. Besides $ZrO_2$ reinforced PDMS, flexible ceramic green bodies can also be fabricated via DIW 3D printing of photocurable acrylate-based ceramic inks[38]. Post-printing shape reconfiguration could proceed through self-assembly-assisted shaping and mold-assisted shaping methods. These reshaped green bodies could be converted to ceramics after sintering.

Different from DIW 3D printing technology that forms 3D structures by extruding viscous inks, digital light processing (DLP) 3D printing turns liquid photocurable resin to solid 3D objects through projection of ultraviolet (UV) patterns onto the surface of resin to trigger localized photopolymerization in combination with the movement of printing stage in vertical direction[13,39]. DLP 3D printing could create more complex and fine 3D structures with the feature size as small as submicron level[40]. However, DLP-based 4D printing of ceramics has not yet been achieved. This is mainly because it lacks (i) the photocurable ceramic resin to form ceramic elastomer with great stretchability; (ii) the photocurable driving materials that enables self-shaping without external loading; (iii) DLP-based multimaterial 3D printing capability that can seamlessly integrate ceramic green body and driving material into one printed structure.

In this work, we report a feasible and efficient manufacturing and design approach to realize direct 4D printing of ceramics. We develop highly photocurable ceramic elastomer slurry and acrylic acid hydrogel precursor with low viscosity for DLP 3D printing. The printed ceramic elastomer green body is highly stretchable and capable of withstanding a tensile strain of up to 700%. The hydrogel serves as driving material, which exhibits a significant dehydration-induced volumetric shrinkage of 65% along with 40 times increase in modulus. Multimaterial DLP 3D printing technology is used to create patterned hydrogel-ceramic laminates where the hydrogel and ceramic elastomer layers form strong interfacial bonding. The flat patterned laminates evolve into complex 3D structures driven by hydrogel dehydration. After debinding and sintering at high temperatures, the evolved 3D structures turn into pure ceramic structures. In order to guide the design of hydrogel-ceramic laminates, we first develop a theoretical model to precisely capture dehydration-induced volumetric shrinkage and modulus increase of the hydrogel, and then implement this model into the Euler-Bernoulli beam theory to generate a design map that builds the relation between the bending curvature and structural parameters of the laminate. The experimental comparison indicates that the sintering leads to the curvature retraction of bent laminate. Through experimental investigations and finite element (FE) simulations, we attribute the curvature retraction to the non-uniform shrinkage in the thickness direction of the laminate during sintering. We further modify the design map by taking account of the curvature retraction, and finally combine the modified design map with FE simulations to build an inverse design flow to determine the structural parameters which make printed flat patterns evolve to target 3D ceramic shapes. Compared with mold-assisted reshaping and manual folding, the hydrogel dehydration-driven direct 4D printing enables simpler and more efficient manufacturing of complex 3D ceramic objects.

## Results

### General process for direct 4D printing of ceramics

As illustrated in Fig. 1a, we start the printing of hydrogel-ceramic laminates on a self-built multimaterial DLP 3D printing system which adopts "bottom-up" projection approach where the UV light engine is placed below the printing platform, and irradiates digitalized UV patterns towards the printing platform that moves vertically to control the thickness of each slice[13,41]. Between the UV light engine and printing platform, there is a glass plate that supports hydrogel precursor and ceramic elastomer slurry containers, and moves horizontally to deliver a needed liquid for the corresponding layer. Details on the 3D printing system can be found in our previous report[13]. The hydrogel precursor mainly consists of 15 wt.% of acrylic acid, 15 wt.% of poly(ethylene glycol) diacrylate (PEGDA), and 70 wt.% of water. Therefore, the hydrogel is named as acrylic acid-PEGDA (AP) hydrogel. The ceramic elastomer slurry is composed of benzyl acrylate (BA), PEGDA and zirconia ($ZrO_2$) nanopowders. The mass ratio of BA-PEGDA resin to $ZrO_2$ ceramic powders is 1: 4. The PEGDA is used as crosslinker, and its content relative to BA-PEGDA resin is adjustable in the range of 1-10 wt.%. We use the ceramic slurry with PEGDA content of 10 wt.% for direct 4D printing of ceramics. Preparation method and uniformity characterization of ceramic slurry can be found in the Methods and Supplementary Information (Supplementary Note 1 and Supplementary Fig. 1). Figure 1b depicts the process that a printed flat hydrogel-ceramic laminate evolves into a curved pure ceramic beam: dehydration-induced shrinkage of the AP hydrogel layer leads to the bending of the laminate; the debinding process at 550 °C removes the hydrogel layer as well as the organic part in the ceramic elastomer layer, while the curved shape is maintained; finally, the sintering process at 1450 °C promotes the grain growth and densification of ceramic parts. We further demonstrate the concept of direct 4D printing ceramics by printing a ceramic flower where the six petals are covered by the hydrogel layers (Fig. 1c). As presented in Fig. 1d, the dehydration process leads to bending of the six bilayer petals covered by hydrogel, while the three petals without hydrogel remain flat. During the debinding process in argon (Ar), all the organics in the hydrogel and elastomer parts decompose into small molecules escaping to the outside, while some carbon species remain in the ceramic layer so that the flower turns to black color. After debinding, the ceramic part has a slightly shrinkage. The sintering process in air removes the residual carbons so that the flower turns to white color again, and the volume shrinks significantly. The scanning electron microscope (SEM) images show that the sintering process makes the ceramic particles grow larger and more densely packed. Details on the debinding and sintering processes as well as the volume change of laminate can be found in Supplementary Notes 2-4 and Supplementary Figs. 2-4. To better demonstrate the dehydration-induced bending as well as the debinding and sintering processes, as shown in Fig. 1e, we print a flat bauhinia flower pattern where the ceramic substrate is covered by a yellow layer of hydrogel. As the time lapses, the flat pattern evolves into a 3D bauhinia flower shape. After the debinding and sintering processes, all organics are removed so that the flower scales down but its 3D shape is maintained.

### Basic properties of ceramic elastomer and AP hydrogel

Figure 2a shows the rheological properties of ceramic elastomer slurry and AP hydrogel precursor. Ceramic elastomer slurry behaves like non-Newtonian liquid and exhibits shear thinning property. As the shear rate increases from $10^{-1}s^{-1}$ to $10^3 s^{-1}$, the viscosity decreases from 5.18 Pa·s to 0.22 Pa·s. The AP hydrogel precursor behaves like Newtonian liquid, and its viscosity is independent of shear rate and always keeps at ~0.05 Pa·s. Figure 2b shows the photorheological properties of ceramic elastomer slurry and AP hydrogel precursor. When UV light is turned on, the hydrogel precursor begins to polymerize rapidly which is reflected by the sharp rise of both storage modulus and loss modulus. After ~10 s of UV irradiation (power: 2.4 mW/cm²), the storage modulus increases by ~5 orders of magnitude, and the loss modulus increases by ~3 orders of magnitude. The gel point where the storage modulus curve intersects the loss modulus curve is identified at 2 s. For the ceramic elastomer slurry, the rapid polymerization reaction begins after 7 s of UV irradiation, and the gel point is identified at 12 s. After 30 s of UV irradiation, the storage modulus increases by ~6 orders of magnitude, and the loss modulus increases by ~4 orders of

magnitude. Figure 2a, b confirm that both ceramic elastomer slurry and AP hydrogel precursor exhibit low viscosity and high photo-reactivity, and thus are suitable for DLP 3D printing.

Figure 2c presents the nonlinear stress-strain behavior of ceramic elastomer which is highly tunable by adjusting the content of cross-linker PEGDA. The decrease in PEGDA leads to the decrease in the modulus of ceramic elastomer, but increase in its stretchability. For example, the ceramic elastomer with 10 wt.% PEGDA has the modulus of 18 MPa and the fracture strain of 40%, while the ceramic elastomer with 1% PEGDA can be stretched by 700% with a modulus of 10 MPa. Specific data about the modulus and fracture strain of ceramic elastomer with different PEGDA contents could be found in Supplementary Note 5 and Supplementary Fig. 5. As shown in Fig. 2d, we print a highly stretchable lattice structure to demonstrate the high printability and stretchability of the ceramic elastomer. The stretched ceramic elastomer structure eventually turns into a pure ceramic structure after debinding and sintering.

Figure 2e shows the stress-strain curves of AP hydrogel with different water contents. As the water content decreases, the tensile modulus and strength of the hydrogel increase, but the fracture strain decreases. As shown in Fig. 2f, the dehydration process (the water content decreases from 70% to 10%) leads to a significant volumetric shrinkage by up to 65% as well as a remarkable Young's modulus increase from 0.27 MPa to 11.16 MPa. In Fig. 2g, we also conduct 180° peeling tests to investigate the interfacial bonding between the ceramic elastomer and AP hydrogel. The interfacial toughness between the two materials is ~16 J/m$^2$ (N/m), which ensures that the ceramic elastomer and AP hydrogel can be tightly bonded during the DLP-based multimaterial 3D printing.

## Dehydration-induced deformation of hydrogel-ceramic laminates

A precise mapping relation between the printed flat pattern and target 3D shape after hydrogel dehydration plays the key role in the design of direct 4D printing of ceramics. As shown in Fig. 3a, in order to quantitatively characterize the dehydration-induced shape change, we print a series of hydrogel-ceramic laminates with variable design parameters, including hydrogel thickness $H_{hg}$, ceramic elastomer thickness $H_{ce}$, and total laminate thickness $H$ ($H = H_{hg} + H_{ce}$). Figure 3b presents the curvature variation of a hydrogel-ceramic laminate ($H_{hg} = 1$ mm, $H_{ce} = 0.8$ mm) during the dehydration process. When the water content in the AP hydrogel decreases from 70 wt.% to 10 wt.%, the laminate changes from a flat shape to almost a circle with curvature $\kappa_d$ of 0.194 mm$^{-1}$. The relationship between the water content and bending curvature over time can be found in Supplementary Note 6 and Supplementary Fig. 6.

In order to guide the structural design, we develop a theoretical model that is able to quantitatively describe the effect of design parameters (i.e. $H_{hg}$, $H_{ce}$ and $H$) on bending curvature of hydrogel-ceramic laminate. As shown in Fig. 2f, the hydrogel dehydration leads

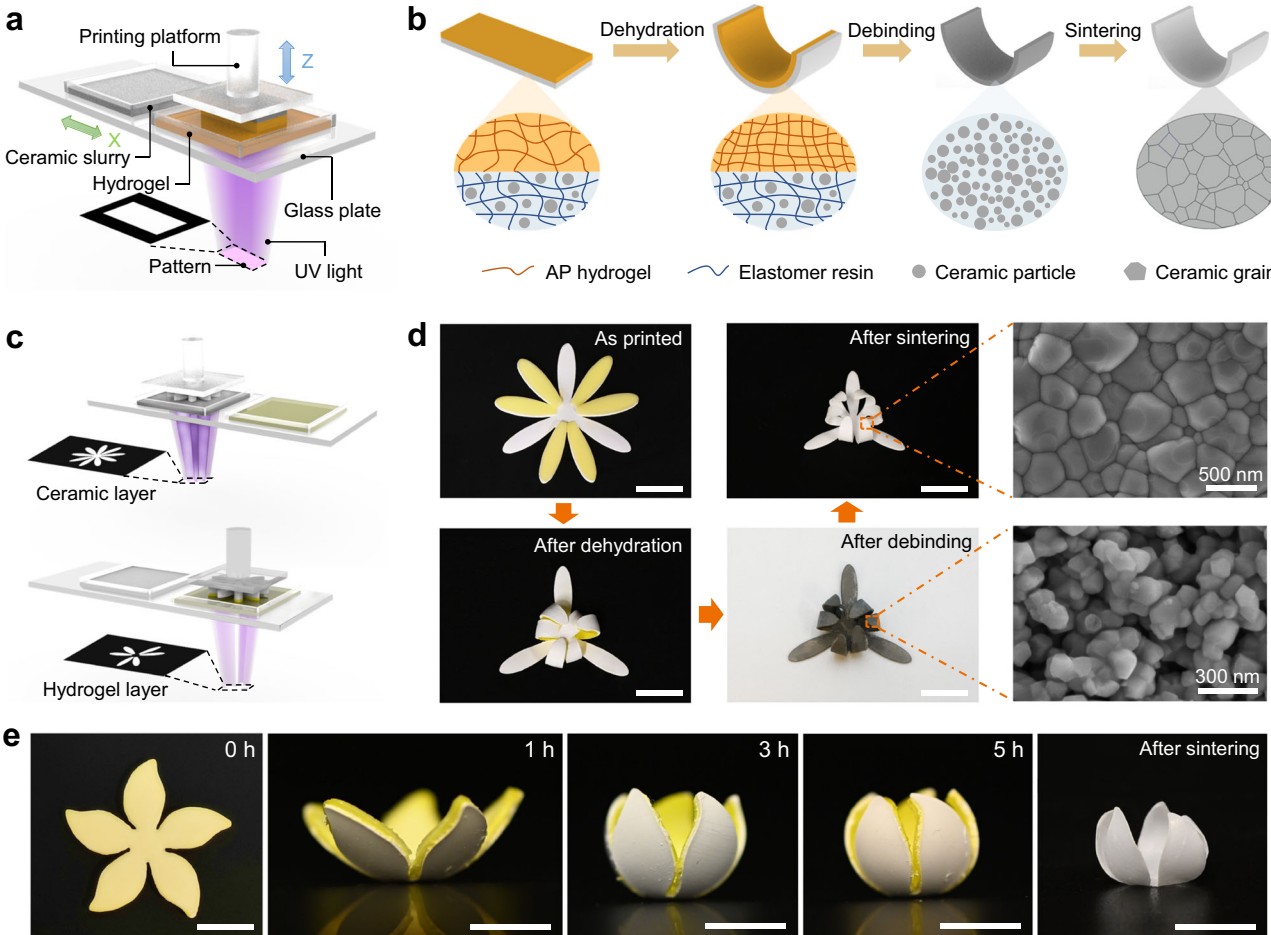

**Fig. 1 | General process of direct 4D printing ceramics. a** Illustration of multi-material 3D printing a hydrogel-ceramic laminate. **b** Processes to convert a hydrogel-ceramic laminate to a pure curved ceramic beam. The upper orange layer and lower grey layer in the laminate represent the AP hydrogel and ceramic elastomer, respectively. **c** Illustration of multimaterial 3D printing a flat flower laminate. After the ceramic layer is printed, the hydrogel vat will move below the printing platform to print hydrogel layer. **d** Demonstration of the processes that convert a printed flat flower laminate to a pure 3D ceramic flower. **e** Evolution process of flower shape over time. Scale bars, 10 mm.

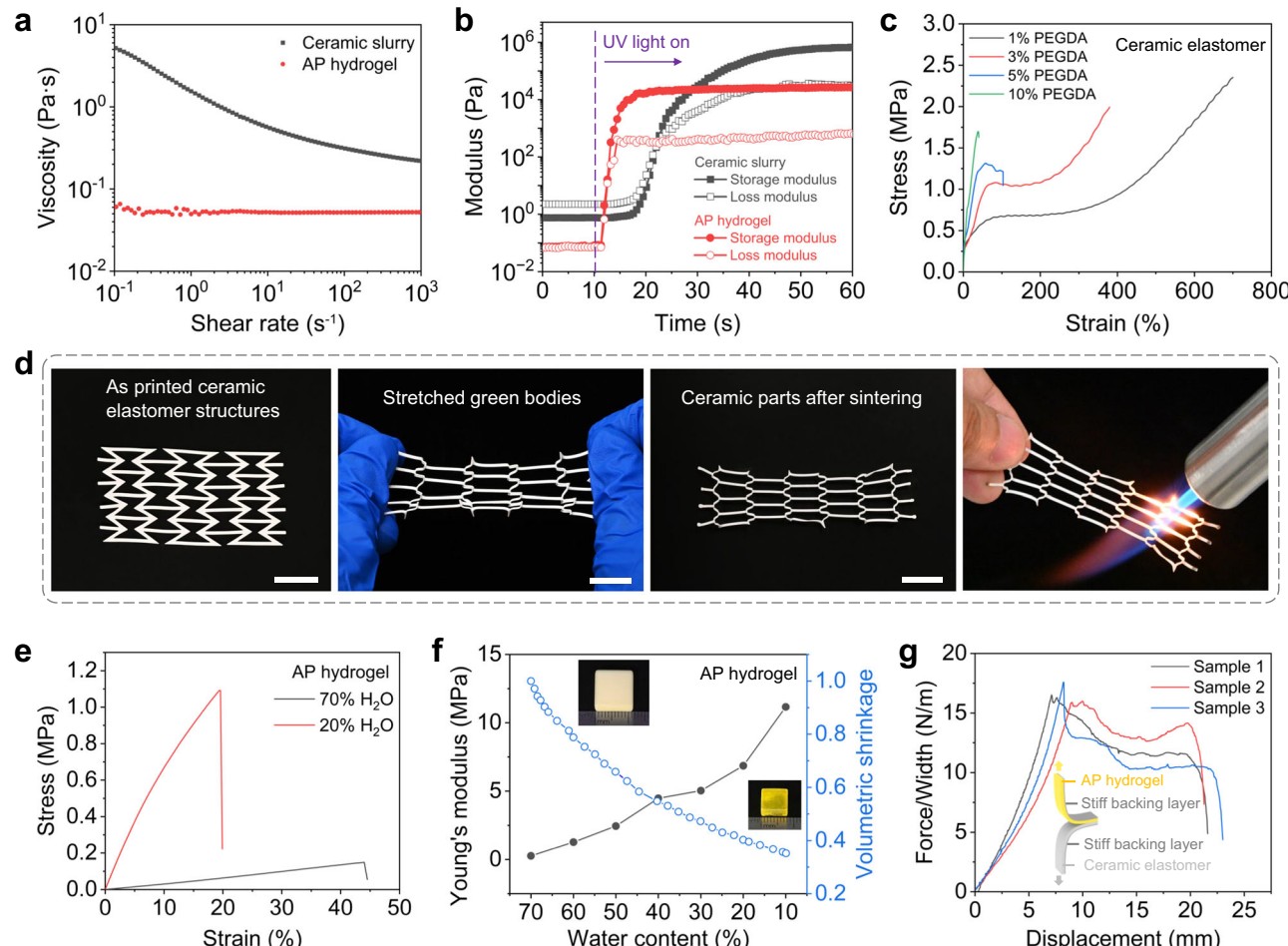

**Fig. 2 | Characterization of material properties. a** Viscosity of ceramic elastomer slurry and acrylic acid- poly(ethylene glycol) diacrylate (AP) hydrogel precursor as a function of shear rate. **b** Storage modulus and loss modulus of ceramic elastomer slurry and AP hydrogel precursor as a function of time during UV irradiation. **c** Stress-strain curves of ceramic elastomer with different poly(ethylene glycol) diacrylate (PEGDA) contents. **d** Demonstration of the stretchability and reshaping ability of the printed ceramic elastomer structures. Scale bars, 15 mm. **e** Stress-strain curves of AP hydrogel with different water contents. **f** Young's modulus and volumetric shrinkage of AP hydrogel as a function of water content during dehydration process. **g** Force/width-displacement curves of peeling test to investigate the interfacial adhesion between ceramic elastomer and AP hydrogel. Source data for Fig. 2a-c and Fig. 2e-g are provided as a Source Data file.

to the coupled effect of volumetric shrinkage and modulus increase of the AP hydrogel. Herein, we adopt the phase evolution based phenomenological model to capture this coupled effect[22]. We assume the dehydration occurs from time 0 to $t_d$ during which the water content $C(t)$ varies from $C_0$ to $C_d$ (based on Fig. 2f, $C_0 = 0.7$ and $C_d = 0.1$). Within a time increment $dt$, the water content decreases by $dC$ which leads to the evolution of a "dry" phase that contributes to the modulus increase of AP hydrogel by $dE_{hg}$. Therefore, the modulus of the AP hydrogel is water content-dependent, which can be described as $E_{hg}[C(t)] = \int_0^{t_d} \frac{dE_{hg}}{dC} \frac{dC}{dt} dt$. Dehydration also results in volumetric change $V(t) = [1 + \varepsilon_d(t)]^3$, where the $\varepsilon_d(t)$ is the isotropic shrinkage strain, and $\varepsilon_d(t) \leq 0$. Both $V(t)$ and $\varepsilon_d(t)$ are water content-dependent, so we have $V(t) = V[C(t)]$ and $\varepsilon_d(t) = \varepsilon_d[C(t)]$. When the dehydration occurs from time 0 to $t_d$, external loads or constraints may be exerted on the hydrogel to yield a time-dependent mechanical strain $\varepsilon_m(t)$, and the total strain on the hydrogel is $\varepsilon_{hg}(t) = \varepsilon_m + \varepsilon_d$. The total stress on the hydrogel at time $t = t_d$ can be written in an integral form:

$$\sigma_{hg} = \int_0^{t_d} \int_t^{t_d} \left[ \frac{dE_{hg}}{dt} \left( \frac{d\varepsilon_{hg}}{d\tau} - \frac{d\varepsilon_d}{d\tau} \right) \right] d\tau dt. \tag{1}$$

Details about the derivation and the incremental form of Eq. (1) can be found in Supplementary Note 7 and Supplementary Fig. 7. Since the strain on the bent hydrogel-ceramic laminate is small (~10% as shown in Supplementary Note 8 and Supplementary Fig. 8), it is legitimate to calculate the stress on the ceramic elastomer as $\sigma_{ce} = E_{ce}\varepsilon_{ce}$, where $E_{ce}$ and $\varepsilon_{ce}$ are Young's modulus and strain of the ceramic elastomer.

During dehydration-induced bending, the hydrogel-ceramic laminate can be modeled as a Euler-Bernoulli beam (Supplementary Note 9 and Supplementary Fig. 9) where the strain $\varepsilon(y)$ along the thickness direction follows:

$$\varepsilon(y) = \varepsilon_0 + \kappa y, \tag{2}$$

where $\varepsilon_0$ is the strain at the hydrogel-ceramic interface ($y = 0$) and $\kappa$ is the bending curvature of the laminate. Thus, the strain on the AP hydrogel along the thickness direction is $\varepsilon_{hg}(y) = \varepsilon_0 + \kappa y$ with $y \in [-H_{hg}, 0]$, so that the total force $F_{hg}$ and total moment $M_{hg}$ applied to the AP hydrogel layer can be calculated as:

$$F_{hg} = \int_{-H_{hg}}^0 \int_0^{t_d} \int_t^{t_d} \left\{ \frac{dE_{hg}}{dt} \left[ \frac{d(\varepsilon_0 + \kappa y)}{d\tau} - \frac{d\varepsilon_d}{d\tau} \right] \right\} \cdot w \cdot d\tau dt dy, \tag{3a}$$

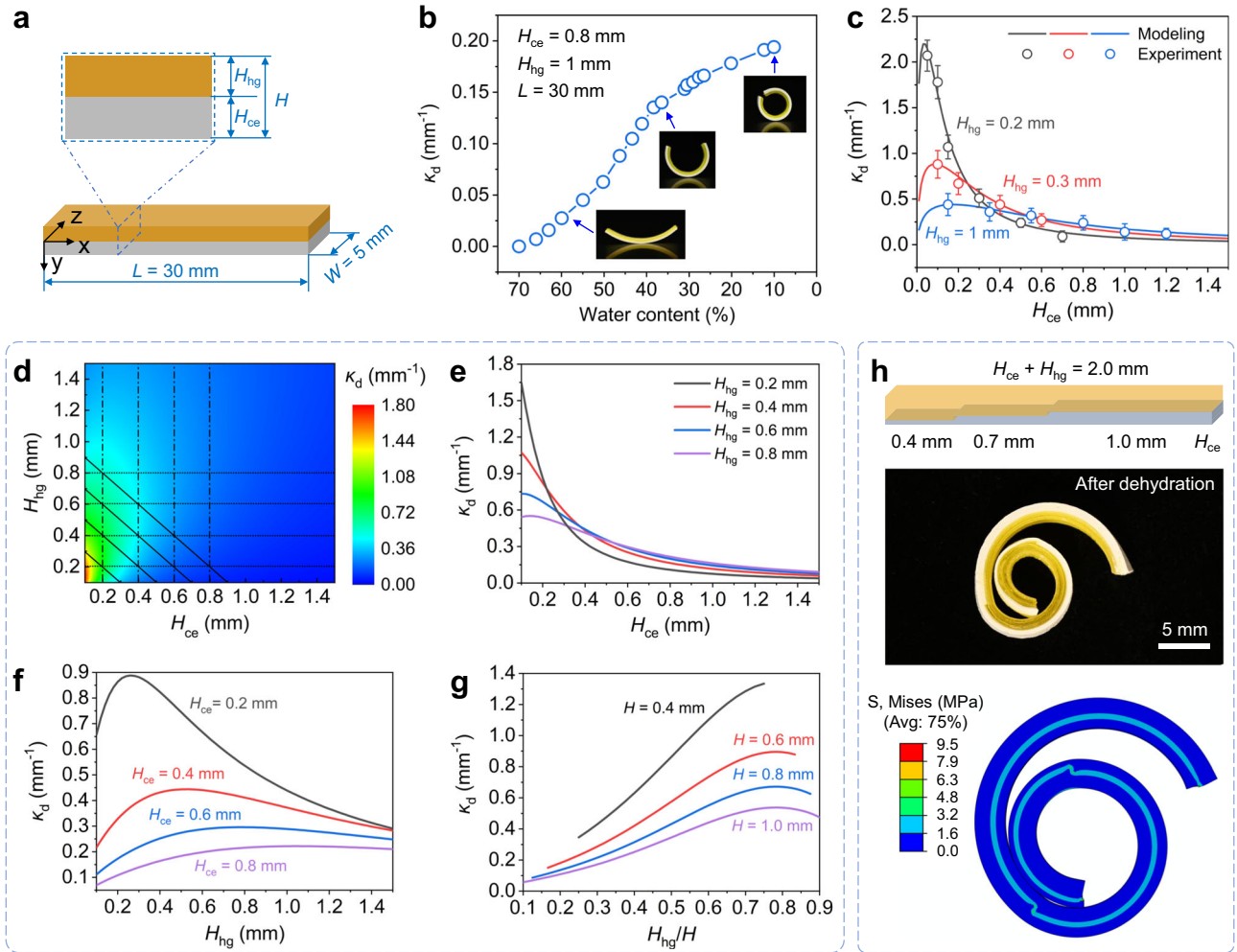

**Fig. 3 | Bending characterizations of the hydrogel-ceramic laminates.**
**a** Illustration of basic structural parameters of hydrogel-ceramic laminates.
**b** Bending curvature of laminates as a function of water content during the hydrogel dehydration process. The insets show the bent laminates with different curvatures. **c** Experiment and modeling results of bending curvature with fixed hydrogel thickness ($H_{hg}$ = 0.2 mm, 0.3 mm, 1 mm) and variable elastomer thickness $H_{ce}$. The error bars represent the standard deviation of curvature. Each curvature value is obtained by measuring 10 samples. **d** Design contour for the bending curvature of bilayer beams with arbitrary thickness arrangement. **e**–**g** Calculated bending curvatures by keeping $H_{hg}$ (**e**), $H_{ce}$ (**f**), and $H$ (**g**) constant while changing other parameters. **h** Coiled bending resulted from the dehydration of a long laminate consisting of three segments with different structural parameters. Source data for Fig. 3b-g are provided as a Source Data file.

$$M_{hg} = \int_{-H_{hg}}^{0} \int_{0}^{t_d} \int_{t}^{t_d} \left\{ \frac{dE_{hg}}{dt} \left[ \frac{d(\varepsilon_0 + \kappa y)}{d\tau} - \frac{d\varepsilon_d}{d\tau} \right] \right\} \cdot w \cdot y \cdot d\tau dt dy, \quad (3b)$$

where $w$ as shown in Fig. 3a is the width of the laminate. The strain on the ceramic elastomer along the thickness direction is $\varepsilon_{ce}(y) = \varepsilon_0 + \kappa y$ with $y \in [0, H_{ce}]$, so that the total force $F_{ce}$ and total moment $M_{ce}$ applied to the ceramic elastomer layer can be calculated as:

$$F_{ce} = \int_{0}^{H_{ce}} \int_{0}^{t_d} \left\{ E_{ce} \cdot \left[ \frac{d(\varepsilon_0 + \kappa y)}{dt} \right] \right\} \cdot w \cdot dt dy, \quad (4a)$$

$$M_{ce} = \int_{0}^{H_{ce}} \int_{0}^{t_d} \left\{ E_{ce} \cdot \left[ \frac{d(\varepsilon_0 + \kappa y)}{dt} \right] \right\} \cdot w \cdot y \cdot dt dy. \quad (4b)$$

In Eqs. (3a)–(4b), there are only two unknowns $\varepsilon_0$ and $\kappa$ which can be calculated by solving two equations according to the condition that dehydration-induced bending is free-standing, and the total force $F_{total}$ and total moment $M_{total}$ applied to the

hydrogel-ceramic laminate are zero:

$$F_{total} = F_{hg} + F_{ce} = 0, \text{ and } M_{total} = M_{hg} + M_{ce} = 0. \quad (5)$$

Details on the derivation of the theoretical model can be found in Supplementary Note 9.

Figure 3c presents the prediction results from the theoretical model by changing $H_{ce}$ from 0.01 mm to 1.5 mm while keeping $H_{hg}$ at 0.2 mm, 0.3 mm and 1 mm respectively. We also carry out experiments to validate the model prediction results, and both results agree well. The model predictions reveal that for the laminate with a fixed $H_{hg}$, its bending curvature first increases with rising $H_{ce}$ until it reaches the maximum bending curvature. The maximum bending curvature increases with the decrease in $H_{hg}$. Figure 3d shows the design map generated from the theoretical model where the bending curvature for each combination of $H_{hg}$ and $H_{ce}$ can be found. From Fig. 3d, the bending curvatures can be predicted by keeping $H_{hg}$ (Fig. 3e), $H_{ce}$ (Fig. 3f), or $H$ (Fig. 3g) constant while changing other parameters. To simulate the dehydration-induced bending in 3D cases, we further implement the theoretical model into the commercial finite element (FE) software packages ABAQUS (Dassault Systems, Johnston, RI, USA)

by writing the user material subroutine (UMAT) to describe the dehydration-induced volumetric shrinkage and modulus increase of the AP hydrogel. Details on the UMAT writing can be found in Supplementary Note 10. As demonstrated in Fig. 3h, we perform FE simulation to predict the bending of a long hydrogel-ceramic laminate consisting of three segments where $H_{ce}$ for each segment is 0.4 mm, 0.7 mm and 1 mm while $H$ is 2 mm for all the three segments. The dehydration induces the long laminate to exhibit coiled bending as $H_{ce}/H$ increases in the three segments. The FE simulation well captures this coiling bending. In addition, although no external force is applied to the laminate, FE simulation reveals that the internal stress is generated at the interface between hydrogel and ceramic elastomer after dehydration.

## Sintering-induced shape retraction of hydrogel-ceramic laminates

The bent hydrogel-ceramic laminates can be later transformed into pure ceramics after the debinding and sintering processes. The experimental comparison shows that the bending angle (or curvature) of the part retracts after the debinding and sintering processes. As shown in Fig. 4a, a dehydrated laminate has curvature of 0.1134 mm$^{-1}$ and bending angle of 207.80°. However, the curvature and bending angle of the ceramic counterpart retract to 0.1040 mm$^{-1}$ and 149.24° respectively (Fig. 4b). To quantitatively characterize the shape retraction, we define the shape retention ratio $R$ which can be calculated as $R = \theta_s/\theta_d$ with initial bending angle $\theta_d$ after dehydration and the final bending angle $\theta_s$ after sintering. To investigate the relation between shape retention ratio $R$ and the initial bending angle $\theta_d$, we print a series of laminates with different thickness of ceramic layer while keeping the hydrogel layer thickness as 1 mm. The result is shown in Fig. 4c where shape retention ratio $R$ stays at ~69.9% for the dehydrated laminates whose bending angles vary from 170° to 330°.

Figure 4d illustrates our hypothesis on the sintering-induced shape retraction. Dehydration leads to the bending of the hydrogel-ceramic laminate where the strain in the ceramic elastomer layer is non-uniform along the thickness direction. Based on Eq. (2), the x-direction normal strain in the ceramic elastomer layer can be expressed as $\varepsilon_{ce}(y) = \varepsilon_0 + \kappa y$ with $y \in [0, H_{ce}]$, and linearly increases along the thickness direction. This non-uniformly distributed strain results in the non-uniform shrinkage during sintering process which eventually causes the shape retraction. In Fig. 4e–h, we conducted FE simulations and experiments to verify this hypothesis. Figure 4e presents the FE analysis (details can be seen in the Methods) indicating that the normal strain $\varepsilon_n$ varies with thickness, and the strain difference $\Delta\varepsilon_n$ between the inner and outer layers of the bent ceramic elastomer is 15%. To confirm the influence of the exerted strain on the sintering-induced shrinkage, we conducted a series of experiments as illustrated in Fig. 4f. For an unstretched ceramic elastomer sample, its length shrinks from $L_0$ to $L_s$ after sintering. Thus, the sintering shrinkage ratio $\eta$ can be defined as $\eta = (L_0 - L_s)/L_0$. For a prestretched sample, its length is first stretched from $L_0$ to $L'_0$ so that the prestretching strain $\varepsilon_{pre}$ can be calculated as $\varepsilon_{pre} = (L'_0 - L_0)/L_0$. Then, the sintering causes the sample to shrink from $L'_0$ to $L'_s$, and the shrinkage ratio $\eta'$ is $(L'_0 - L'_s)/L'_0$. Thus, the shrinkage difference $\Delta\eta$ is $\eta' - \eta$. Figure 4g shows that the sintering shrinkage ratio $\eta$ increases from 21.74% to 26.20% by increasing the prestretching strain $\varepsilon_{pre}$ from 0 to 25%. We further use FE simulation to confirm the non-uniform shrinkage does result in the shape retraction. As shown in Fig. 4h, the uniform shrinkage merely causes the arc to scale down without shape change. In contrast, the arc opens due to the non-uniform shrinkage. Therefore, the non-uniform normal strain along the thickness direction is a crucial factor leading to the shape retraction during sintering process.

Based on the above hypothesis, we can modify the design map in Fig. 3d by taking account of the shape retraction after sintering. According to Eq. (2), we can calculate the normal strain difference $\Delta\varepsilon_n$

between the outer and inner surfaces of ceramic elastomer layer, and $\Delta\varepsilon_n$ equals to $\kappa_d H_{ce}$. Since Fig. 3d already shows the relation between $\kappa_d$ and arbitrary combination of $H_{hg}$ and $H_{ce}$, we can build the relation between $\Delta\varepsilon_n$ and $H_{hg}$-$H_{ce}$ combination (Fig. 4i). Because the normal strain difference $\Delta\varepsilon_n$ causes the shape retraction, we employ the following exponential function $\Delta\eta = C_1 \exp(\Delta\varepsilon_n/C_2) + C_3$ to establish the mapping relation between sintering-induced shrinkage difference $\Delta\eta$ and normal strain difference $\Delta\varepsilon_n$. The constants $C_1$, $C_2$ and $C_3$ can be determined from the experimental results presented in Supplementary Fig. 10. Then, we modify the curved beam model[42] to build the relation between the curvature after dehydration $\kappa_d$ and the one after sintering $\kappa_s$ as

$$\kappa_s = \kappa_d - \frac{3\Delta\eta}{2H_{ce}}. \qquad (6)$$

Details on the modification of the curved model can be found in Supplementary Note 11. Figure 4j shows the contour of $\kappa_s$ with arbitrary combination of $H_{hg}$ and $H_{ce}$ in the range of $0.1 - 1.5$ mm. At last, based on the definition of shape retention ratio $R = \theta_s/\theta_d$, we rewrite $R$ by using the calculated curvatures $\kappa_d$ and $\kappa_s$ as $R = \kappa_s(1 - \eta_0)/\kappa_d \times 100\%$, where $\eta_0$ denotes the sintering-induced shrinkage of ceramic elastomer without prestretch. As shown in Fig. 4k, the predicted shape retention ratio $R$ lies in the vicinity of 70% for the laminate samples with fixed hydrogel thickness $H_{hg} = 1.0$ mm and elevated ceramic elastomer thickness $H_{ce}$ ranging from 0.4 to 1.2 mm, which agrees with the experimental results (Fig. 4c) very well.

## Design flow for direct 4D printing of ceramics

Figure 5a presents the design flow for direct 4D printing of ceramics, which includes five steps: 3D modeling, flat pattern design, theoretical model calculation, FE simulation, and experiment. We take the fabrication of a thin-walled regular tetrahedron as an example to illustrate the detailed design and manufacturing process of 4D printing ceramics. As shown in Fig. 5b, we first extract the key feature parameters from the target object. In the case of a regular tetrahedron, the side length $L_s$ is 10 mm, and the calculated dihedral angle $\theta_s$ is 70.5°. Then, we design geometric parameters for the flat pattern that can be obtained by unfolding each face of the regular tetrahedron into to the same plane. As shown in Fig. 5c, all the faces only consist of ceramic elastomer. The bottom face and side faces are connected by three hinges composed of hydrogel-ceramic laminates. Considering the shrinkage of ceramics after sintering, the side length $L_d$ of each triangular face is calculated to be 12.78 mm ($L_d = L_s/\eta = 12.78$ mm). The hinge length is equal to $L_d$, and the width is set to be 5 mm. The undetermined parameters are hydrogel thickness $H_{hg}$ and hinge thickness $H$ which can be calculated through the developed theoretical model. Figure 5d presents the relation between bending angle $\theta_s$ and $H_{hg}/H$. The dihedral angle of the sintered ceramic tetrahedron can reach 70.5° when $H = 1$ mm and $H_{hg}/H = 0.69$. Based on the structural parameters determined in Fig. 5c, d, we perform FE simulations to predict the shape transformation from the flat 2D pattern to the dehydrated and sintered 3D objects. As shown in Fig. 5e, the final sintered 3D shape is basically consistent with our initial design. Finally, Fig. 5f presents the experiment in which a ceramic tetrahedron is directly 4D printed. We use multimaterial DLP 3D printing to fabricate the 2D flat pattern with the determined design parameters, transform the 2D pattern to a 3D tetrahedron through dehydration, and convert the polymer-based tetrahedron to a ceramic one by sintering. The sintering process leads to the volume shrinkage and shape retraction, which is well predicted by the FE simulation.

## Direct 4D printing of ceramics with complex structures

The direct 4D printing approach for ceramics developed in this work offers great flexibility in design and manufacturing to create a wide

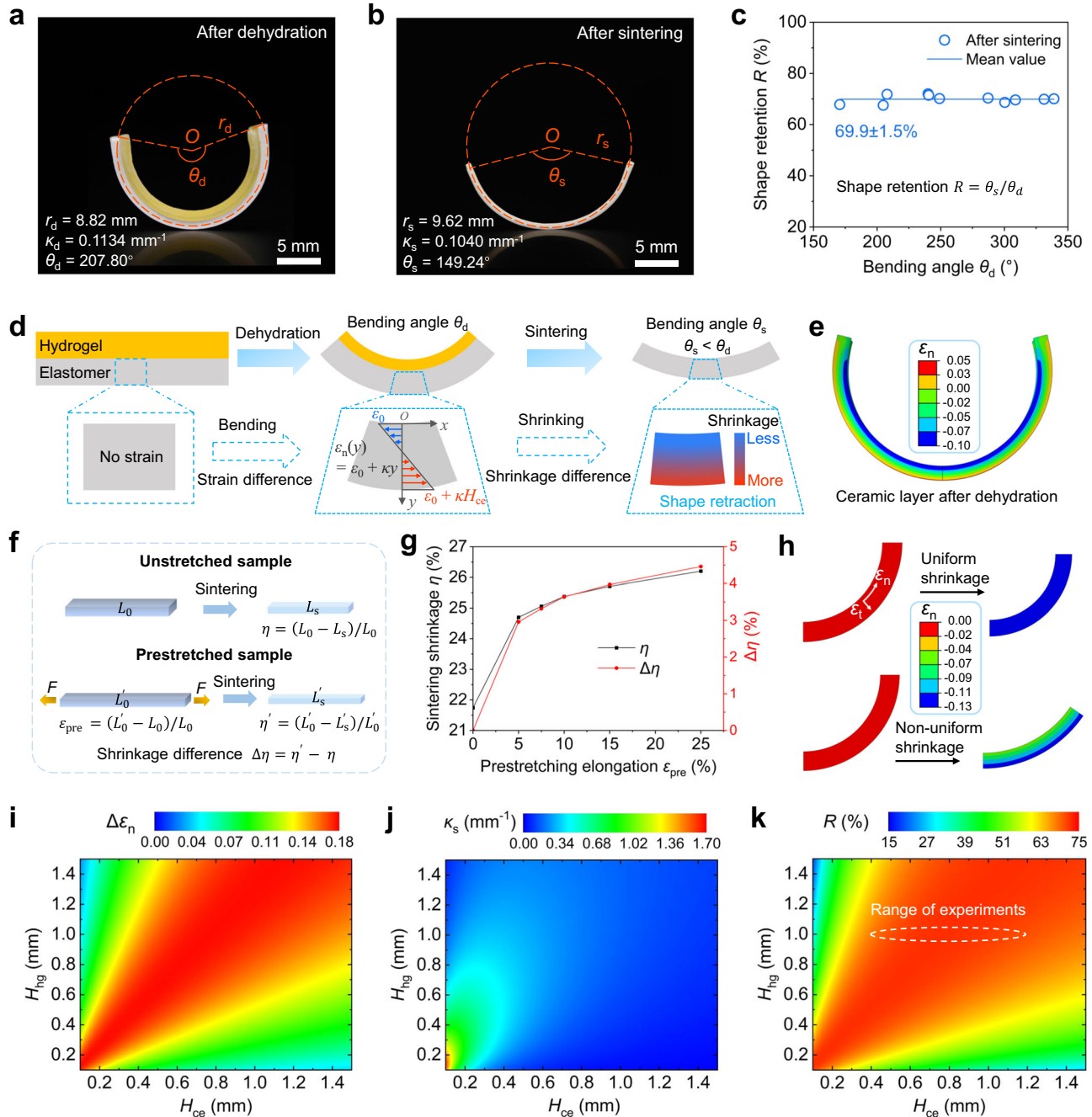

**Fig. 4 | Characterization of shape change after sintering. a** The shape of hydrogel-ceramic laminate ($H_{hg}$ = 1.0 mm, $H_{ce}$ = 0.94 mm) after dehydration. **b** The bent ceramic beam after sintering. **c** The shape retention $R$ of ceramic beams with different initial bending angles $\theta_d$. The error is the standard deviation of all data. **d** Illustration of shape change in the dehydration and sintering processes. **e** FE simulation result of strain contour of ceramic layer after dehydration. **f** Illustration of shrinkage difference of unstretched and prestretched ceramic elastomers. **g** The sintering shrinkage of ceramic elastomers with different prestretching elongations. **h** FE simulation of shape change under uniform shrinkage and non-uniform shrinkage. **i**–**k** Strain difference contour in longitudinal direction (**i**), curvature contour after sintering (**j**), and shape retention ratio contour (**k**), with arbitrary hydrogel thickness $H_{hg}$ and ceramic elastomer thickness $H_{ce}$ in the range of 0.1 mm-1.5 mm. Source data for Fig. 4c, g, i-k are provided as a Source Data file.

variety of 3D ceramic structures evolved from different flat patterns, which is reflected by the five demonstrations as shown in Fig. 6. Figure 6a presents a self-assembled cube from a flat pattern. We adopt the design method similar to that of the regular tetrahedron in Fig. 5c, and set hydrogel-ceramic laminates as hinges between two adjacent square faces. Dehydration drives the hinges to bend into a target angle, and folds the flat pattern into the desired cube. Figure 6b, c show 4D printing process of ceramic Miura origami and crane. Different from Fig. 6a, there are hydrogel-ceramic laminate hinges located in both

side of the flat patterns, and each part of the pattern bends towards different directions. By elaborately designing the parameters and locations of hinges, we can precisely control the bending angle and direction of each part, and achieve more complex configurations. Figure 6d demonstrates the three-blade fan with spiral blades which are twisted by dehydration of the equally spaced inclined hydrogel fibers printed on the ceramic elastomer matrix. Different from the overall bending shown in Fig. 6a, the dehydrated bilayer laminate undergoes a helical transformation, where the direction of the inclined

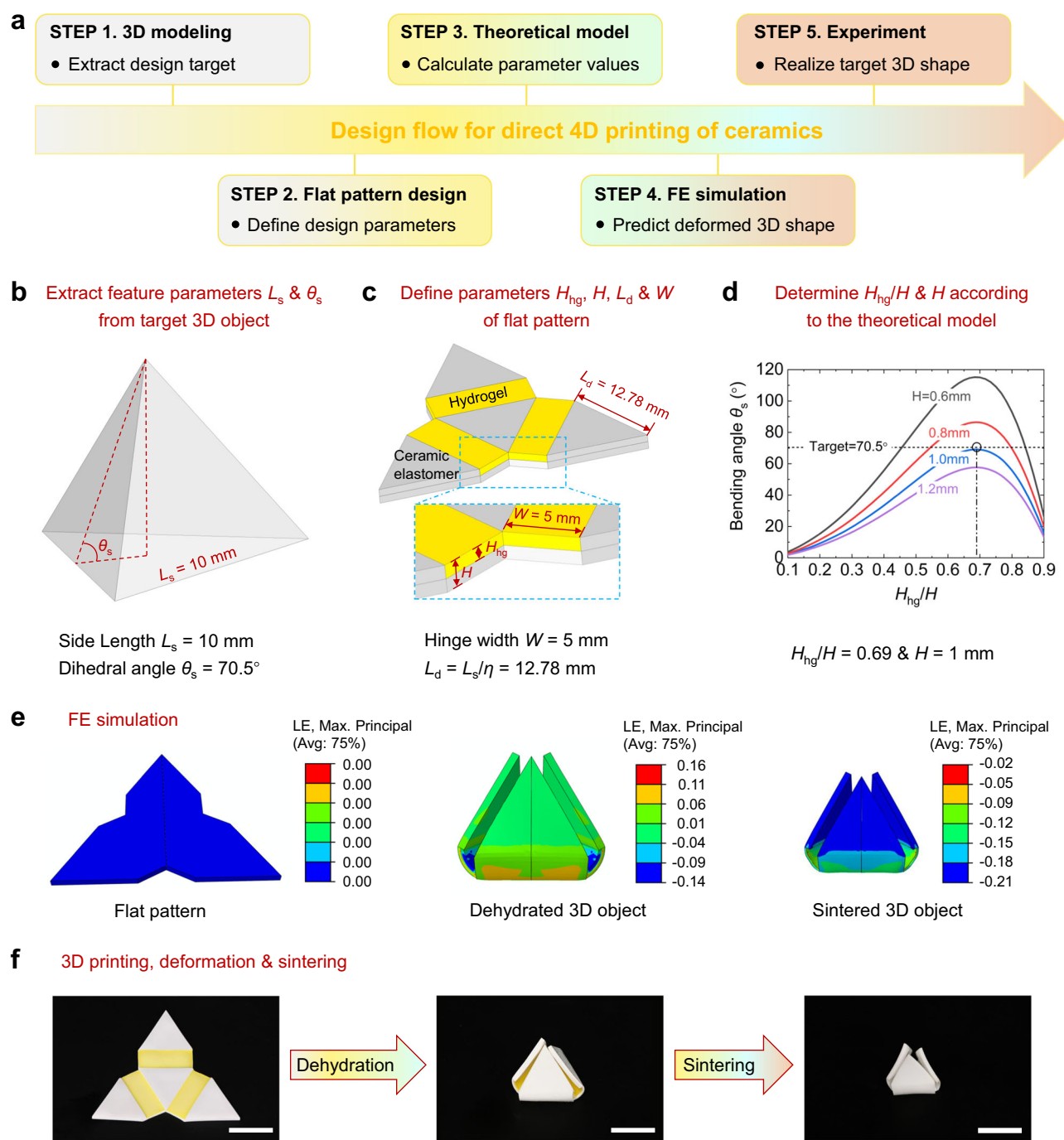

**Fig. 5 | Design flow for direct 4D printing of ceramics. a** Five main steps for 4D printing ceramics. **b–f** Detailed design and manufacturing process of 4D printing ceramics taking the fabrication of a thin-walled regular tetrahedron as an example: extract feature parameters of 3D tetrahedron (**b**), design flat pattern and define parameters (**c**), determine specific parameter values according to the theoretical model (**d**), finite element (FE) simulation to predict deformed shape (**e**), print the flat pattern and obtain the target 3D object through hydrogel dehydration-induced deformation and sintering process (**f**). Scale bars, 10 mm. Source data for Fig. 5d are provided as a Source Data file.

hydrogel fiber serves as a new local bending axis for the separated hydrogel-ceramic segments. Figure 6e demonstrates the direct 4D printing of ceramic scorpion. We construct hydrogel-ceramic laminates located at the claws, tail and feet of the scorpion. The hydrogel layers at the claws and tail are above the ceramic elastomer layers, making the two claws raise up and the tail curl up. The hydrogel layers at the feet are below the ceramic elastomer layers, causing the eight feet to bend downwards. It is worth mentioning that the width of the scorpion feet is only less than 1 mm, indicating that our 4D printing ceramic method is also suitable for fabrication of small-sized objects.

Details on the design parameters for the 2D patterns can be found in Supplementary Figs. 11-15.

## Discussion

In this work, we report a feasible and efficient manufacturing and design approach to realize direct 4D printing of ceramics. We develop highly photocurable ceramic elastomer slurry and AP hydrogel precursor suitable for DLP 3D printing. The BA-PEGDA-based ceramic elastomer is highly stretchable with an elongation at break up to 700%. The AP hydrogel serves as driving material, undergoing a significant volumetric

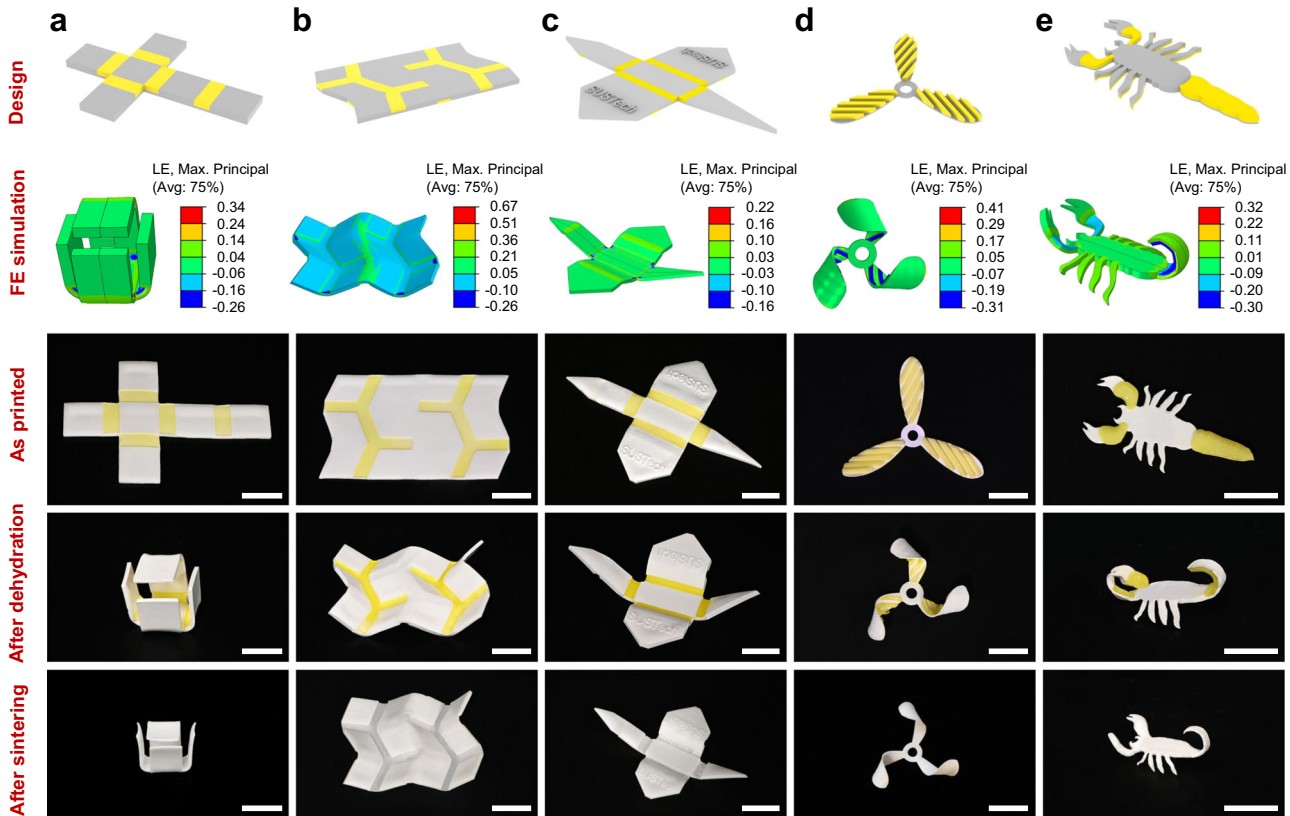

**Fig. 6 | Representative demonstrations of direct 4D printing of ceramics. a–c** Regular cube (**a**), Miura origami (**b**) and crane (**c**) evolved from flat patterns to 3D structures. The hydrogel-ceramic laminates are used as folding hinges while the non-bending segments consist of ceramic elastomer only. **d** The three-blade fan with twisted blades evolved from flat pattern with inclined hydrogel fibers on ceramic elastomer. Scale bars in (**a–d**), 10 mm. **e** The scorpion with bent claws, feet and curly tail evolved from flat pattern. Scale bars in (**e**), 5 mm.

shrinkage of 65% and a 40-fold increase in modulus when dehydrated. Multimaterial DLP 3D printing technology is used to fabricate patterned hydrogel-ceramic laminates that could evolve into complex 3D structures driven by hydrogel dehydration. After debinding and sintering at high temperatures, the evolved 3D structures are converted to pure ceramics. To facilitate the design of hydrogel-ceramic laminates, we develop a theoretical model to precisely capture dehydration-induced volumetric shrinkage and modulus increase of the hydrogel, and then implement this model into the Euler-Bernoulli beam theory to generate design map that builds a relationship between the bending curvature and structural parameters of the laminate. Aiming at the sintering-induced shape retraction of bent laminate, we conduct experimental investigations and FE simulations, attributing the curvature retraction to the non-uniform shrinkage in the thickness direction of the laminate during sintering. Then, we modify the design map by taking account of the curvature retraction. Combining the modified design map and FE simulations, we establish an inverse design flow to determine the structural parameters which make printed flat patterns evolve to target 3D ceramic shapes. Finally, we use five cases including cube, Miura origami, crane, three-blade fan and scorpion, to demonstrate the capacity to create complex 3D structures. Through deploying the hydrogel-ceramic laminates at different positions in the flat pattern via multimaterial DLP 3D printing technology, we can achieve local deformations such as bending, twisting, folding and origami precisely, and then realize a wide variety of complex shape configuration. The proposed direct 4D printing approach for ceramics can be applied to various ceramic materials. The high efficiency and great flexibility of this approach in producing complex 3D objects will break through the current limitations of ceramic structure design, opening up new avenues for the application of ceramic materials in a broader range of engineering fields.

## Methods

### Preparation of ceramic elastomer slurry and hydrogel precursor

The photocurable ceramic elastomer slurry was first prepared by mixing $ZrO_2$ ceramic powders (6.08 g/cm³, 3Y-TZP, $d_{50} = 0.56$ μm, Shenzhen Adventuretech Co., Ltd., China), benzyl acrylate (BA, Bide Pharmatech Ltd., China), poly(ethylene glycol) diacrylate (PEGDA, Mw = 700, Innochem Technology Co., Ltd., China), KOS110 (Guangzhou Kangoushuang Trade Co., Ltd., China) and diphenyl(2,4,6-trimethylbenzoyl) phosphine oxide (TPO, Bide Pharmatech Ltd., China). In the mixture, KOS110 (5 wt.% of $ZrO_2$) was used as the dispersant and TPO (0.5 wt.% of resin) was used as the photoinitiator. Then the mixture was ball-milled for 8 h at 540 rpm (F-P400E, Hunan Focucy Experimental Instrument Co., Ltd., China) to obtain homogeneous $ZrO_2$ slurry. In this work, we use a slurry with solid content of 40 vol% for printing.

The AP hydrogel precursor was prepared by mixing the acrylic acid (Shanghai Aladdin Bio-Chem Technology Co., LTD, China), PEGDA (Mw = 700), self-developed water-soluble photoinitiator TPO[43] and deionized water. The acrylic acid and PEGDA were mixed in a ratio of 1:1. The water-soluble TPO content was 5 wt.% of acrylic acid-PEGDA mixture. The water content of the precursor used for 4D printing was 70 wt.%. To improve the printing accuracy, we added 0.005% Food Yellow NO.4 into the AP hydrogel precursor.

### Multimaterial 3D printing, dehydration and heat treatment processes

We printed all the hydrogel-ceramic laminates on a self-built DLP-based multimaterial 3D printing apparatus with the layer thickness of 100 μm. The wavelength of UV light is 405 nm. We set the energy dose as 60 mJ/cm² for the printing of ceramic layers and 20 mJ/cm² for

hydrogel layers. After printing, the hydrogel-ceramic laminates were dehydrated at room temperature (25 °C) for 7 h and then at 80 °C in an oven for 4 h. In the dehydration process, the laminates underwent bending deformation due to the large volume shrinkage of the hydrogel. The curved samples were heated at 550 °C for 4 h in Ar for debinding, and then sintered at 1450 °C in air to obtain the ceramic parts. A tube furnace (GSL-1700X, Hefei Kejing, China) and a muffle furnace (KSL-1700X, Hefei Kejing, China) with a maximum service temperature of 1700 °C were used to perform the debinding and sintering processes, respectively. After debinding and sintering, the ceramic has a linear shrinkage ratio of 3% and 21.74%, respectively. Details about the debinding and sintering processes could be found in the Supplementary Information.

## Materials characterization

The viscosities of ceramic elastomer slurry and AP hydrogel precursor were measured at room temperature using a rheometer (Discovery HR-3, TA Instruments, USA) equipped with parallel plates (diameter of 20 mm and gap size of 200 μm) across a shear rate of 0.1-1000 s$^{-1}$. A UV light source (405 nm, 2.4 mW/cm$^2$) was attached to the rheometer to perform photorheology experiments. The storage modulus and loss modulus were measured to characterize the gel transition of ceramic slurry and AP hydrogel precursor under UV irradiation.

Uniaxial tensile experiments of ceramic elastomers with different PEGDA contents were performed using MTS machine (Model E45, MTS Systems Corporation, USA) equipped with a 100 N load cell under a crosshead speed of 5 mm/min at room temperature. The mechanical properties of AP hydrogel with different water contents were measured by a DMA machine (Discovery DMA850, TA Instruments, USA). The samples for 180° peeling test were printed with dimensions of 60 mm × 10 mm × 2 mm (1 mm for each layer) and a 10 mm long precrack. Stiff backing layers were attached to the hydrogel layer and ceramic elastomer layer to constrain their deformation. The 180° peeling test was performed on the MTS machine with a 100 N load cell at speed of 10 mm/min to investigate the interfacial bonding between AP hydrogel and ceramic elastomer.

To obtain the relation between water content and volume change of the hydrogel during dehydration, we printed hydrogel block with dimensions of 12 mm × 12 mm × 6 mm, and measured its mass and dimensions every 15 min at room temperature. After 7 h at room temperature, the rate of water loss became very slowly. Then we placed it in an 80 °C oven to continue the dehydration process. After dried at 80 °C for 4 h, the mass and dimensions no longer changed. The water content and volume could be calculated according to the measured mass and dimensions.

The dehydrated laminates were photographed by a camera (Nikon Z7II). The bending curvatures of the bilayer laminates were measured by using the image analysis software ImageJ.

Morphological observations were conducted on a field emission scanning electron microscope (Apreo2 S Lovac, Thermo Fisher Scientific, US) with a 5 kV accelerating voltage. Prior to SEM observation, samples were sputtered with platinum for 30 s using a vacuum coater (Leica EM ACE200, Leica Microsystems, Germany).

## FE simulation

A phase-evolution-based constitutive model that incorporates the dehydration-induced volume shrinkage and material stiffening was implemented into an ABAQUS (Dassault Systems, Johnston, RI, USA) user material subroutine (UMAT) (Supplementary Note 10) to simulate the mechanical behavior of the AP hydrogel[22]. Material parameters in the UMAT was determined by fitting the experimental results (Fig. 2f) that depict the effects of water content on dehydration strain and Young's modulus. In simulation, water content $C$ was mimicked by temperature and coupled temperature-displacement analysis was performed with C3D8T elements in ABAQUS element library. Hydrogel

and ceramic elastomer parts were assumed to be perfectly bonded. All the structural geometry and boundary conditions were kept to be consistent with actual experiments. Upon simulating the sintering of ceramic elastomer, element deletion technique was employed to remove the hydrogel layer.

## Data availability

The data generated in this study are provided in the Source Data file. The dataset for this work is available at Figshare, reference[44]. Source data are provided with this paper.

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

## Acknowledgements

Q.G. acknowledges the National Natural Science Foundation of China (No. 12072142), the Key Talent Recruitment Program of Guangdong Province (No. 2019QN01Z438), and the support by the Science, Technology and Innovation Commission of Shenzhen Municipality under grant no. ZDSYS20210623092005017. C.Y. acknowledges the National Natural Science Foundation of China (No. 12102329).

## Author contributions

R.W., C.Y. and J.C. contributed equally. R.W., C.Y., J.C. and Q.G. conceived the ideas and designed the research; C.Y. and Q.G. performed the theoretical calculation and FE simulation. R.W., J.C., X.H. and H.Y. investigated the material properties. R.W. and J.C. printed the structures. B.J., H.L. and J.B. participated in the analysis of experimental data. R.W., C.Y. and Q.G. drafted the manuscript; C.Y. and Q.G. revised the manuscript. Q.G. supervised the project.

## Competing interests

The authors declare no competing interests.
