## [Peer Review File · Nature Communications]

REVIEWER COMMENTS

Reviewer #1 (Remarks to the Author):

The difficulties associated with 4D printing ceramics stem from their intrinsic inflexibility. In contrast to more flexible substances, the process of attaining 4D printing capabilities with ceramics is complex as a result of their limited capacity for deformation. The proposed methodology involves a two-step process, wherein a photocurable ceramic elastomer slurry and a hydrogel precursor are utilized to fabricate hydrogel-ceramic laminates through multimaterial 3D printing. The process of dehydrating hydrogel results in the formation of intricate three-dimensional structures, which subsequently undergo maturation into ceramics via the process of sintering. A design workflow has been developed to facilitate the direct 4D printing of a wide range of ceramic objects. This innovation presents a potentially effective approach to furthering the development of ceramic 4D printing technology by overcoming the inherent constraints of the material. The work is timely and interesting, however the following should be addressed:

1- Multimaterial Printing: Incorporating multiple materials or functional elements within the ceramic matrix can add complexity to the printing and transformation process. How this is optimised in the work.

2- Explain why DLP_ is best choice than DIW?

3- How structural integrity of the system is maintained before sintering to get the final desired product.

4- How uniform and homogenous resin is guaranteed?

5- How such application of ceramic hydrogel could have applications in biomedical and vascularised parts in 4D, discuss from the works in "4D Printing for Vascular Tissue Engineering: Progress and Challenges" and "3D-Printed Phase-Change Artificial Muscles with Autonomous Vibration Control"

6- How the maximum temperature is achieved to be 1450C and how could be optimised and what are the challenges?

7- In the manuscript explain what the main factors are affecting the shrinking and bending of the structure before sintering. How this factors, during the drying are measured over the time. The similar example in the 4D modeling in could be referenced "Multimaterial 4D printing with a tunable bending model". However, this work experiences an hygroscopic phenomenon in morphing that should be elaborated. This part is so important to be elaborated.

Reviewer #2 (Remarks to the Author):

In this manuscript, the authors report an approach to realize direct 4D printing of ceramics driven by hydrogel dehydration. The authors have developed a theoretical model and proposed a design flow to guide the structural parameter design of hydrogel-ceramic elastomer flat patterns accurately. This work is innovative and shows excellent experiment effects. The presented cases demonstrate the great potential of the approach. This work has achieved DLP-based ceramic 4D printing through non-contact active deformation, which is of great significance for 4D printing of ceramics. Thus, it is recommended for publication in Nature Communications.

My questions and suggestions are listed as follows:

1. Why do the modelling results in Fig. 3c show that the variation of curvature with ceramic elastomer thickness is not monotonic?
2. The title of the horizontal axis ("Prestretching elongation") in Fig. 4g covered the numbers partially. The author should modify this graph.
3. How long does the dehydration process take before debinding and sintering in high temperature?

4. Polyacrylamide hydrogel is another commonly used hydrogel. Can the AP hydrogel in this work be replaced by polyacrylamide hydrogel?

Reviewer #3 (Remarks to the Author):

This manuscript titled "Direct 4D printing of ceramics driven by hydrogel dehydration" by Dr. Yuan and co-authors report a feasible and efficient manufacturing and design approach to realize direct 4D printing of ceramics. The results showed that 3D ceramic with a wide variety of complex shape configuration can be achieved by using the proposed direct 4D printing approach. Overall, the paper is well written and figures are well organized. Therefore, I would like to recommend publication after addressing minor issues.

Major questions:

Lines 180: The statement about the peeling tests is not clear. Rewrite is required. Meanwhile, if the material selection will affect the interlaminar deformation compatibility of the laminate during dehydration?

Lines 199: Is the ductility of ceramic elastomer layer affect the deformation curvature as well? In Fig.3 (b), the surface stress will increase with the increase of curvature and thickness, which I think may limit the engineering application of this method.

Lines 268: Please provide more details about ABAQUS model, such as the relationship between temperature and water content, and interface contact property.

Minor points:

-Lines 118: What is the mass fraction of PEGDA in the ceramic slurry, and how it is determined in the later experiments?

Lines 119: Does the volume of ceramic elastomer layer change during dehydration and debinding?

-Typos:

Lines 129 and 501: "Ar2" to "Ar"

RESPONSE TO REVIEWERS' COMMENTS

Reviewer #1 (Remarks to the Author):

The difficulties associated with 4D printing ceramics stem from their intrinsic inflexibility. In contrast to more flexible substances, the process of attaining 4D printing capabilities with ceramics is complex as a result of their limited capacity for deformation. The proposed methodology involves a two-step process, wherein a photocurable ceramic elastomer slurry and a hydrogel precursor are utilized to fabricate hydrogel-ceramic laminates through multimaterial 3D printing. The process of dehydrating hydrogel results in the formation of intricate three-dimensional structures, which subsequently undergo maturation into ceramics via the process of sintering. A design workflow has been developed to facilitate the direct 4D printing of a wide range of ceramic objects. This innovation presents a potentially effective approach to furthering the development of ceramic 4D printing technology by overcoming the inherent constraints of the material. The work is timely and interesting.

Response: We thank the reviewer for these positive comments on our work.

However, the following should be addressed:

Comment 1.1: Multimaterial Printing: Incorporating multiple materials or functional elements within the ceramic matrix can add complexity to the printing and transformation process. How this is optimised in the work.

Response: We thank the reviewer for raising this question. In this work, we report a feasible and efficient manufacturing and design approach to realize direct 4D printing of ceramics. We develop highly photocurable ceramic elastomer slurry and acrylic acid hydrogel precursor with low viscosity for DLP 3D printing. Multimaterial DLP 3D printing technology is used to create patterned hydrogel-ceramic laminates where the hydrogel and ceramic elastomer layers form strong interface bonding. The flat patterned laminates evolve into complex 3D structures driven by hydrogel dehydration. After debinding and sintering at high temperatures, the evolved 3D structures turn into pure ceramic structures.

Our work is not only the optimization of the multimaterial 3D printing and transformation processes, but also the **inverse design process** for direct 4D printing of ceramics driven by hydrogel dehydration. The design flow for direct 4D printing of ceramics is shown in **Figure R1 (Figure 5 in the manuscript)**.

This work advances the manufacturing and design approach for 4D printing of ceramics in the following five aspects:

- (i) **Photocurable ceramic elastomer with high stretchability and photocurable hydrogel with large dehydration-induced volumetric shrinkage.** We develop highly photocurable ceramic elastomer and acrylic acid hydrogel precursor solutions with low viscosity suitable for DLP 3D printing. The printed ceramic elastomer green body is highly stretchable and capable of withstanding a tensile strain of up to 700%. The hydrogel serves as driving material, which exhibits a significant dehydration-induced volumetric shrinkage of 65% along with 40 times increase in modulus. The ceramic elastomer could form strong interface bonding with hydrogel.
- (ii) **High-precision multimaterial DLP 3D printing technology.** Compared with extrusion-based DIW technology that previously reported 4D printing of ceramics has adopted, DLP 3D printing technology has higher printing resolution. In this work, we create high-precision patterned hydrogel-ceramic laminates via multimaterial DLP 3D printing technology.
- (iii) **Non-contact active deformation driven by hydrogel dehydration.** The hydrogel could drive the flat laminates to evolve into 3D structures accurately upon dehydration. After debinding and sintering, pure ceramic part with well-retained shape could be obtained. Compared with mold-assisted reshaping and manual folding, the hydrogel dehydration driven direct 4D printing enables simpler and more efficient manufacturing of complex 3D ceramic objects.
- (iv) **High-fidelity theoretical model to assist structural parameter design.** We develop a theoretical mechanics model that considers dehydration-induced deformation and sintering-induced shape retraction to construct design map to guide the structural parameter design.
- (v) **Design flow for direct 4D printing of ceramics.** Combining the design map and FE simulations, we establish an inverse design flow to determine the structural parameters which make printed flat patterns evolve to targeted 3D ceramic shapes. We use five cases including cube, Miura origami, crane, three-blade fan and scorpion, to demonstrate the capacity of the proposed approach to create various complex 3D ceramic objects efficiently.

Figure R1 (Figure 5 in the manuscript). Design flow for direct 4D printing of ceramics. **a**, Five main steps for 4D printing ceramics. **b-f**, Detailed design and manufacturing process of 4D printing ceramics taking the fabrication of a thin-walled regular tetrahedron as an example: extract feature parameters of 3D tetrahedron (**b**), design flat pattern and define parameters (**c**), determine specific parameter values according to the theoretical model (**d**), FE simulation to predict deformed shape (**e**), print the flat pattern and obtain the target 3D object through hydrogel dehydration-induced deformation and sintering process (**f**). Scale bars, 10 mm.

Comment 1.2: Explain why DLP is best choice than DIW?

Response: We thank the reviewer for raising this question. DIW is an extrusion-based 3D printing technology, which deposits viscoelastic inks with shear-thinning property

to build 3D objects. The printing resolution of DIW technology depends on the diameter of nozzles and rheological properties of inks. Currently, research on ceramic 4D printing is mainly based on DIW technology. We have summarized recent advances of DIW-based 4D printing of ceramics in the second paragraph of the Introduction section of manuscript.

DLP 3D printing performs the localized photocuring through projecting 2D UV patterns on the surface of liquid polymer resin. DLP combines the feature of high resolution with fast speed, and can create more complex and fine 3D structures with the feature size as small as submicron level. However, DLP-based 4D printing of ceramics has not yet been achieved. This is mainly because it lacks (i) the photocurable ceramic resin to form ceramic elastomer with great stretchability; (ii) the photocurable driving materials that enables self-shaping without external loading; (iii) DLP-based multimaterial 3D printing capability that can seamlessly integrate ceramic green body and driving material into one printed structure. This work primarily aims to achieve direct 4D printing of ceramics based on multimaterial DLP printing from four aspects: materials development, multimaterial 3D printing technology, theoretical model establishment, and structural design approach. We have highlighted the advantages of DLP technology in the revised manuscript.

In the beginning the third paragraph, we have briefly discussed the difference between DIW and DLP 3D printing:

“Different from DIW 3D printing technology that forms 3D structures by extruding viscous inks, digital light processing (DLP) 3D printing turns liquid photocurable resin to solid 3D objects through projection of ultraviolet (UV) patterns onto the surface of resin to trigger localized photopolymerization in combination with the movement of printing stage in vertical direction^{13,39}. DLP 3D printing could create more complex and fine 3D structures with the feature size as small as submicron level⁴⁰.”

Comment 1.3: How structural integrity of the system is maintained before sintering to get the final desired product.

Response: We thank the reviewer for raising this question. The AP hydrogel and ceramic elastomer we developed in this work have good interfacial strength (**Figure R2a, Fig. 2g in the manuscript**), which ensures that the hydrogel-ceramic elastomer laminates can maintain the integrity of the structure during the dehydration and deformation processes without interface peeling (**Figure R2b**). After debinding in argon, the hydrogel is removed along with the organics in ceramic elastomer, and the ceramic part remains (**Figure R2b**).

In order to develop an appropriate debinding process, we conducted Thermogravimetry Analysis (TGA) experiment. TGA (STA 449 F3, NETZSCH, Germany) was carried out in nitrogen with flow rate of 50 mL/min at a heating rate of 10 °C/min. **Figure R2c (Supplementary Figure 2a)** shows the Thermogravimetry-Derivative Thermogravimetry (TG-DTG) curves of ceramic elastomer and dehydrated AP hydrogel. The TG curves indicate that the organics in ceramic elastomer and AP hydrogel mainly decompose at 300 °C-500 °C. The DTG curves indicate that the organics in ceramic elastomer and AP hydrogel decompose most violently at 398.3 °C and 423.8 °C, respectively. Based on these results, we formulate the debinding process with multi-step heating strategy as shown in **Figure R2d (Supplementary Figure 2b)**. The debinding process was carried out in argon using a tube furnace. When the temperature was lower than 300 °C, a relatively fast heating rate of 1 °C/min was adopted to save time. Then the samples were heated up to 400 °C for 2 h, 425 °C for 2 h, and 550 °C for 4 h, respectively. The heating rate was set as 0.25 °C/min. This process ensured that the gas produced by the decomposition of organics could slowly escape from the inside of material without cracking or destroying the structure. After 4 h at 550 °C, the debinding process was basically completed. Finally, the temperature was raised to 800 °C and held for 2 h. After that, the structure obtained enough holding strength to keep intact when it was transferred to a muffle furnace for sintering. After sintering, we got the final desired product. We have added the TGA experiments and debinding process in the Supplementary Information (Supplementary Note 2 and Supplementary Figure 2a-c).

Figure R2. **a**, Force/width-displacement curves of peeling test to investigate the interfacial adhesion between ceramic elastomer and AP hydrogel (**Fig. 2g in the manuscript**). **b**, Photographs of hydrogel-ceramic laminate after printing, dehydration and debinding, respectively (**Supplementary Figure 2c**). Scale bars, 5 mm. **c**, TG-DTG curves of ceramic elastomer and dehydrated AP hydrogel (**Supplementary Figure 2a**). **d**, The debinding process in argon (**Supplementary Figure 2b**).

Comment 1.4: How uniform and homogenous resin is guaranteed?

Response: We thank the reviewer for raising this question. The ceramic slurry is composed of photosensitive resin (benzyl acrylate and PEGDA), ceramic powder, dispersant and photoinitiator. The dispersant (KOS 110, Guangzhou Kangoushuang Trade Co., Ltd., China) enables ceramic particles to be better dispersed in the resin. In order to break up agglomerated ceramic particles and ensure that these particles are dispersed uniformly enough in the resin, we use ball milling to mix them. Details about the slurry preparation process have been highlighted in Methods section of the revised manuscript.

“The photocurable ceramic elastomer slurry was first prepared by mixing ZrO₂ ceramic powders (6.08 g/cm³, 3Y-TZP, d₅₀ = 0.56 μm, Shenzhen Adventuretech Co., Ltd.,

China), benzyl acrylate (BA, Bide Pharmatech Ltd., China), poly(ethylene glycol) diacrylate (PEGDA, Mw = 700, Innochem Technology Co., Ltd., China), KOS110 (Guangzhou Kangoushuang Trade Co., Ltd., China) and diphenyl(2,4,6-trimethylbenzoyl) phosphine oxide (TPO, Bide Pharmatech Ltd., China). In the mixture, KOS110 (5 wt.% of ZrO₂) was used as the dispersant and TPO (0.5 wt.% of resin) was used as the photoinitiator. Then the mixture was ball-milled for 8 h at 540 rpm (F-P400E, Hunan Focucy Experimental Instrument Co., Ltd., China) to obtain homogeneous ZrO₂ slurry.”

Figure R3 (Supplementary Figure 1) compares the ceramic slurry prepared using a ball mill with that prepared using a planetary mixer. As shown in **Figure R3a** and **R3c**, we take a droplet of slurry onto a glass sheet and blow the droplet with compressed air to form a film. From the optical images in **Figure R3b** and **R3d**, we can see that the slurry prepared using the planetary mixer contains a large number of agglomerated ceramic particles with a size of tens of microns, while the slurry prepared using ball milling has no obvious particles. Therefore, the ball milling method can guarantee the ceramic slurry uniform and homogenous. We have added the characterization of slurry uniformity in the Supplementary Information (Supplementary Note 1 and Supplementary Figure 1).

Figure R3 (Supplementary Figure 1). Optical images of ceramic slurries prepared by different mixing methods. a,b, Using a planetary mixer. **c,d,** Using a ball mill. Scale bars, 200 μm.

Comment 1.5: How such application of ceramic hydrogel could have applications in biomedical and vascularised parts in 4D, discuss from the works in “4D Printing for Vascular Tissue Engineering: Progress and Challenges” and “3D-Printed Phase-Change Artificial Muscles with Autonomous Vibration Control”.

Response: We thank the reviewer for these valuable references. We have cited these two works in the revised manuscript (Ref. 19 and Ref. 20). The discussion is as follows:

Applications in tissue engineering: 4D-printed ceramic structures have significant potential as scaffolds in tissue engineering. Scaffolds play a fundamental role in tissue engineering by providing essential support and guidance for the regeneration of bone tissue. One crucial aspect is the establishment of a vascular network within the scaffold to supply nutrients and oxygen, facilitating cell migration, proliferation and differentiation during the bone tissue repair process. Thus, vascularization stands as a primary requirement for successful bone tissue regeneration. Bioceramics like hydroxyapatite have emerged as competitive candidates for scaffold materials in bone tissue regeneration due to their special chemical composition, high compressive strength and excellent bioactivity. Incorporating ceramic-hydrogel composite structure through 3D printing not only enhances the scaffold’s mechanical properties but also improves the biological activity. By loading growth factors and micronutrients within the hydrogel, this approach can effectively promote angiogenesis and osteogenesis in the regenerating tissue.

Applications in artificial muscles: Ceramic materials can be combined with hydrogels or other active materials to create composites with specific properties. These composites can be engineered to have high strength and controlled contraction properties, making them suitable for use in artificial muscles. Certain ceramic materials, such as piezoelectric ceramics, change size when subjected to an electric field. This property is exploited in artificial muscles for controlled movements. Piezoelectric ceramics can be used to create muscles that respond to electrical signals, making them valuable in robotics and biomedical engineering applications.

Comment 1.6: How the maximum temperature is achieved to be 1450°C and how could be optimised and what are the challenges?

Response: We thank the reviewer for raising these questions. The samples are sintered in a muffle furnace (KSL-1700X, Hefei Kejing, China) with a maximum service temperature of 1700 °C. The specific sintering process is illustrated in **Figure R4a**

(Supplementary Figure 2d). The temperature is raised to 1450 °C by multi-step heating and held for 2h for sintering. For ZrO₂ ceramic in this work, 1450 °C is the optimized sintering temperature. After sintered at 1450 °C for 2 h, the flexural strength of ZrO₂ ceramic reaches 800 MPa. We have added the sintering process in the Supplementary Information (Supplementary Note 2 and Supplementary Figure 2d).

The optimal sintering temperature is obtained by measuring the three-point bending strength of ceramic specimens after sintered at different temperatures. The specimens which exhibit a maximum flexural strength have the optimal sintering temperature. A series of ceramic specimens were printed and then sintered at different temperatures. We carried out three-point bending test using MTS machine (Model E45, MTS Systems Corporation, USA) equipped with three-point bending fixture. **Figure R4b** shows the illustration of three-point bending test. The loading rate was set as 0.5 mm/min. A typical force-displacement curve is shown in **Figure R4c (Supplementary Figure 3a)**. The following equation is used to calculate the flexural strength σ_b :

$$\sigma_b = \frac{3PL}{2bh^2}$$

where P is the maximum force, L is the support span, b is the width of specimen, and h is the thickness of specimen. **Figure R4d (Supplementary Figure 3b)** shows the flexural strength of ZrO₂ ceramic after sintered at different temperatures. The ceramic sintered at 1450 °C has the highest flexural strength (~800 MPa). Therefore, 1450 °C is the optimal sintering temperature for ZrO₂ ceramic in this work. We have added the three-point bending test in the Supplementary Information (Supplementary Note 3 and Supplementary Figure 3).

Figure R4. Sintering process and three-point bending test. **a**, The sintering process in air (Supplementary Figure 2d). **b**, Illustration of three-point bending test. **c**, Typical force-displacement plot of three-point bending test (Supplementary Figure 3a). **d**, Flexural strength of ZrO₂ ceramic after sintering at different temperatures (Supplementary Figure 3b).

Comment 1.7: In the manuscript explain what the main factors are affecting the shrinking and bending of the structure before sintering. How this factors, during the drying are measured over the time. The similar example in the 4D modeling in could be referenced “Multimaterial 4D printing with a tunable bending model”. However, this work experiences a hygroscopic phenomenon in morphing that should be elaborated. This part is so important to be elaborated.

Response: We thank the reviewer for this suggestion. During the dehydration process, the volume of the hydrogel layer shrinks significantly, while the ceramic elastomer layer only bends and its volume is approximately considered to be unchanged. For AP hydrogel, volumetric shrinkage after dehydration depends on the initial water content. The larger the initial water content, the greater the volumetric shrinkage after dehydration. We printed hydrogel block with dimensions of 12 mm × 12 mm × 6 mm, and measured its mass and dimensions every 15 min at room temperature. After 7 h at room temperature, the rate of water loss became very slowly. Then we placed it in an

80 °C oven to continue the dehydration process. After dried at 80 °C for 4 h, the mass and dimensions no longer changed. The water content and volume can be calculated according to the measured mass and dimensions. Then, we can obtain the relation between water content and volume change of the hydrogel during dehydration, as shown in **Figure R5a (Fig. 2f in the manuscript)**. In this work, we use the hydrogel with water content of 70 wt.% for the 4D printing experiment. When the water content decreases from 70 wt.% to 10 wt.%, the hydrogel has a volumetric shrinkage of 65%. To obtain the mechanical properties of AP hydrogel with different water contents, we prepared a series of AP hydrogel precursors with different water contents. Samples with dimensions of 20 mm × 5 mm × 1 mm were printed. The mechanical properties were measured by a DMA machine (Discovery DMA850, TA Instruments, USA). **Figure R5a (Fig. 2f in the manuscript)** also shows the Young's modulus of AP hydrogel with different water contents. We have added the measuring methods in the Methods section of the revised manuscript.

The curvature of hydrogel-ceramic laminate is related to the volumetric shrinkage and modulus change of the hydrogel during dehydration, the modulus of the ceramic elastomer, as well as the thickness of the hydrogel layer and the ceramic elastomer layer. In this work, we select AP hydrogel with 70 wt.% water content, and BA-PEGDA-based ceramic elastomer. The final curvature of hydrogel-ceramic laminate after dehydration depends on the thickness of the hydrogel layer and the ceramic elastomer layer. **Figure R5b (Fig. 3a in the manuscript)** shows the illustration of basic structural parameters of hydrogel-ceramic laminates. We printed a series of laminates with different H_{ce} and H_{hg} . After dehydration at room temperature for 7 h and at 80 °C for 4 h, we measured their bending curvatures. **Figure R5c (Supplementary Figure 6)** shows the water content and curvature of hydrogel-ceramic laminate as a function of time during dehydration. **Figure R5d (Fig. 3c in the manuscript)** shows the curvature as a function of H_{ce} while keeping H_{hg} at 0.2 mm, 0.3 mm and 1 mm, respectively. These experimental results are basically consistent with the theoretical results. **Figure R6 (Fig. 3d-g in the manuscript)** shows the design contour and the calculated bending curvatures of bilayer laminates with different thickness of hydrogel layer and ceramic elastomer layer. We have added the changes in water content and bending curvature over time in the Supplementary Information (Supplementary Note 6 and Supplementary Figure 6).

Figure R5. The shrinking and bending behaviors during the dehydration. **a**, Young's modulus and volumetric shrinkage of AP hydrogel as a function of water content during the dehydration process (**Fig. 2f in the manuscript**). **b**, Illustration of basic structural parameters of hydrogel-ceramic laminates (**Fig. 3a in the manuscript**). **c**, Water content and curvature of hydrogel-ceramic laminate as a function of time during the dehydration process (**Supplementary Figure 6**). **d**, Experiment and modeling results of bending curvature with fixed hydrogel thickness ($H_{hg} = 0.2 \text{ mm}$, 0.3 mm , 1 mm) and variable elastomer thickness H_{ce} (**Fig. 3c in the manuscript**).

Figure R6 (Fig. 3d-g in the manuscript). Calculation results of bending curvature after dehydration. a, Design contour for the bending curvature of bilayer beams with arbitrary thickness arrangement. **b-d**, Calculated bending curvatures by keeping H_{hg} (b), H_{ce} (c), and H (d) constant while changing other parameters.

About the theoretical modeling: We thank the reviewer for recommending the valuable paper on 4D printing modeling. We have cited the paper in the revised manuscript (Ref. 5). However, our modeling for 4D printing of hydrogel-ceramic laminate mainly has the following four characteristics:

- (i) We develop the constitutive model of hydrogel which can describe the coupled effect of stiffness increase and volume shrinkage in the dehydration process.
- (ii) We implant the constitutive models of hydrogel and elastomer into the Euler-Bernoulli beam theory, and draw the curvature contour of hydrogel-elastomer laminates with different geometric parameter combinations.
- (iii) Based on the observed experimental phenomenon of curvature retraction after sintering, we modify the design model and obtain the relation between the geometric parameter combination and the curvature after sintering.
- (iv) We construct a design flow for direct 4D printing of ceramics, and achieve an

inverse design for 4D printing of ceramics driven by target geometric shapes. The developed model is written into user material subroutine (UMAT). We can use ABAQUS to simulate the entire process of dehydration-driven deformation and sintering-induced shape retraction.

About the hygroscopic phenomenon in morphing: In this work, the deformation of hydrogel-ceramic laminate is driven by hydrogel dehydration. As we described above, the dehydration is a very slow process: the water content only decreases from 70 wt.% to 30 wt.% after the hydrogel is placed in room temperature for 7 hours, and the water content is further decreased to 10 wt.% after the hydrogel is placed in the oven at 80 °C for 4 h. The hygroscopic process is the inverse process of dehydration, but both are the diffusion processes where the water molecules diffuse into (or out of) the hydrogel polymer network. Therefore, the hygroscopic process is also very slow, and the obvious hygroscopic effect can only be observed after hours. In our experiment, after the sample is placed in the oven at 80 °C for 4 h, we immediately fetch it out to measure the curvature, at which point the residual water in the hydrogel is ~10 wt.%. The whole operation process only takes a few minutes which is sufficiently short so that the hygroscopic phenomenon can be neglected. Moreover, the subsequent debinding process would cause the hydrogel to lose all the water until it is completely decomposed. After debinding, only the ceramic part remains. Therefore, the hygroscopic phenomenon does not affect the experimental results.

Reviewer #2 (Remarks to the Author):

In this manuscript, the authors report an approach to realize direct 4D printing of ceramics driven by hydrogel dehydration. The authors have developed a theoretical model and proposed a design flow to guide the structural parameter design of hydrogel-ceramic elastomer flat patterns accurately. This work is innovative and shows excellent experiment effects. The presented cases demonstrate the great potential of the approach. This work has achieved DLP-based ceramic 4D printing through non-contact active deformation, which is of great significance for 4D printing of ceramics. Thus, it is recommended for publication in Nature Communications.

Response: We thank the reviewer for these positive comments on our work.

My questions and suggestions are listed as follows:

Comment 2.1: Why do the modelling results in Fig. 3c show that the variation of curvature with ceramic elastomer thickness is not monotonic?

Response: We thank the reviewer for raising this question. To better understand the non-monotonic curvature variation, we can consider an extreme case where the ceramic elastomer thickness H_{ce} is far less than the hydrogel thickness H_{hg} . In this circumstance, because the stiffness contribution from the ceramic elastomer is neglectable, the hydrogel-ceramic laminate can be ideally treated as an isotropic hydrogel that uniformly shrinks upon dehydration without creating bending curvature. This extreme case indicates that the bending curvature of the bilayer structure will not monotonically decrease with the rising ceramic elastomer thickness.

In fact, bending of the hydrogel-ceramic laminate is cooperatively determined by the strain mismatch and stiffness competing between hydrogel and ceramic elastomer. When the stiffness contribution from one part becomes far less comparable to the other (for instance, extremely thin or extremely thick ceramic elastomer), the bending curvature is negligible because the beam has become nearly isotropic.

Comment 2.2: The title of the horizontal axis ("Prestretching elongation") in Fig. 4g covered the numbers partially. The author should modify this graph.

Response: We thank the reviewer for this suggestion. We have modified the graph in the revised manuscript.

Comment 2.3: How long does the dehydration process take before debinding and sintering in high temperature?

Response: We thank the reviewer for raising this question. The dehydration time is related to the size of the hydrogel. For the hydrogel-ceramic laminate with L of 30 mm, W of 5 mm, H_{hg} of 1 mm, H_{ce} of 0.8 mm), **Figure R7 (Supplementary Figure 6)** shows the changes in water content and bending curvature of the laminate over time during dehydration. First, the laminate is dehydrated at room temperature (25 °C). After 7 h at room temperature, the remaining water content in the hydrogel is ~29% and the bending curvature of laminate is 0.160 mm^{-1} . Then, the laminate is placed in an 80 °C oven to continue the dehydration process. After dried at 80 °C for 4 h, the water content decreases to ~10% and the bending curvature is 0.194 mm^{-1} . We have added this information in the Supplementary Information (Supplementary Note 6 and Supplementary Figure 6).

Figure R7 (Supplementary Figure 6). Water content and curvature of hydrogel-ceramic laminate as a function of time during the dehydration process.

Comment 2.4: Polyacrylamide hydrogel is another commonly used hydrogel. Can the AP hydrogel in this work be replaced by polyacrylamide hydrogel?

Response: We thank the reviewer for raising this question. Polyacrylamide hydrogel cannot replace the AP hydrogel in this work, which is confirmed by our experiment. We have printed polyacrylamide hydrogel-ceramic elastomer laminate with L of 30 mm, W of 5 mm, H_{hg} of 1 mm and H_{ce} of 1 mm. The water content of polyacrylamide hydrogel is also 70 wt.%. The photographs of laminate after dehydration, debinding and sintering are shown in **Figure R8**. After dehydration, the laminate bends into an arch. Unlike AP hydrogel that disappears completely after debinding, polyacrylamide hydrogel expands

substantially, causing the laminate to bend in the opposite direction (**Figure R8b**). After sintering, the ceramic is broken into two parts (**Figure R8c**). Therefore, polyacrylamide hydrogel is not a suitable material used for direct 4D printing of ceramics in this work.

Figure R8. The photographs of polyacrylamide hydrogel-ceramic elastomer laminate (a) after dehydration, (b) after debinding, and (c) after sintering.

Reviewer #3 (Remarks to the Author):

This manuscript titled “Direct 4D printing of ceramics driven by hydrogel dehydration” by Dr. Yuan and co-authors report a feasible and efficient manufacturing and design approach to realize direct 4D printing of ceramics. The results showed that 3D ceramic with a wide variety of complex shape configuration can be achieved by using the proposed direct 4D printing approach.

Overall, the paper is well written and figures are well organized. Therefore, I would like to recommend publication after addressing minor issues.

Response: We thank the reviewer for these positive comments on our work.

Major questions:

Comment 3.1: Lines 180: The statement about the peeling tests is not clear. Rewrite is required. Meanwhile, if the material selection will affect the interlaminar deformation compatibility of the laminate during dehydration?

Response: We thank the reviewer for this valuable suggestion. **Figure R9 (the inset of Figure 2g in the manuscript)** illustrates the 180° peeling test. The samples for 180° peeling test were printed with dimensions of 60 mm×10 mm×2 mm (1 mm for each layer) and a 10 mm long pre-crack. Stiff backing layers were attached to the hydrogel layer and ceramic elastomer layer to constrain their deformation. The 180° peeling test was performed on the MTS machine with a 100 N load cell at speed of 10 mm/min to investigate the interfacial bonding between AP hydrogel and ceramic elastomer. We have rewritten this experiment in the Methods section of the revised manuscript. The modifications are highlighted in yellow.

“The samples for 180° peeling test were printed with dimensions of 60 mm× 10 mm× 2 mm (1 mm for each layer) and a 10 mm long pre-crack. Stiff backing layers were attached to the hydrogel layer and ceramic elastomer layer to constrain their deformation. The 180° peeling test was performed on the MTS machine with a 100 N load cell at speed of 10 mm/min to investigate the interfacial bonding between AP hydrogel and ceramic elastomer.”

The material selection has a great influence on the interlaminar deformation compatibility of the laminate during dehydration. First of all, the interfacial strength of hydrogel and ceramic elastomer should be strong enough to ensure that no delamination occurs during dehydration. Second, the volumetric shrinkage and modulus variation of hydrogel during dehydration affect the bending curvature of the laminate. Finally, the

mechanical properties of ceramic elastomers also have a great influence on the bending curvature of the laminate. Thus, material selection is very critical for direct 4D printing of ceramics.

Figure R9 (The inset in Figure 2g in the manuscript). Illustration of 180° peeling test.

Comment 3.2: Lines 199: Is the ductility of ceramic elastomer layer affect the deformation curvature as well? In Fig.3 (b), the surface stress will increase with the increase of curvature and thickness, which I think may limit the engineering application of this method.

Response: We understand the reviewer's concern about the ductility of ceramic elastomer. However, FE simulation result in **Figure R10a (Supplementary Figure 8a)** indicates that the maximum tensile longitudinal strain (ϵ_n) produced by the ceramic elastomer is less than 10% when the hydrogel-ceramic laminate ($H_{hg} = H_{ce} = 1$ mm) bends into an arc shape that approaches a full circle. Comparison of the stress-strain curve between linear elastic model and experiment in **Figure R10b (Supplementary Figure 8b)** indicates that the ceramic elastomer in this work performs in the linear elastic range.

Secondly, according to our design contour shown in **Figure R11 (Figure 3d in the manuscript)**, a large thickness of ceramic elastomer (H_{ce}) will yield a small bending curvature κ_d of the hydrogel-ceramic laminate upon dehydration. Furthermore, because the internal strain ϵ is linearly proportional to bending curvature κ_d (**Eq. 2 of the manuscript**) and the ceramic elastomer approximately follows the linear elastic constitutive model, we can obtain that a large thickness of ceramic elastomer (H_{ce}) will

generate a small surface stress. Therefore, the current material system for ceramic elastomer is qualified for the desired bending transformation towards potential applications.

Figure R10 (Supplementary Figure 8). Deformation of the bent laminate. a, Strain contour of the hydrogel-ceramic laminate after dehydration. The inner layer is AP hydrogel. The outer layer is ceramic elastomer. **b,** Comparison of the stress-strain curves between linear elastic model and experiment.

Figure R11 (Figure 3d in the manuscript). Design contour for the bending curvature of bilayer beams with arbitrary thickness arrangement.

Comment 3.3: Lines 268: Please provide more details about ABAQUS model, such as the relationship between temperature and water content, and interface contact property.

Response: We thank the reviewer for this suggestion. We have added more detail on FE simulation in the Methods section. We use the relationship among modulus, volume shrinkage and water content (Fig. 2f) to simulate the dehydration behavior of hydrogel.

Because hydrogel dehydration is performed in an isothermal condition, the relationship between temperature and water content is not taken into account in our FE simulation. In addition, because no delamination between hydrogel and ceramic elastomer is observed in our dehydration experiments, we assume the two parts are perfectly bonded in our FE simulations.

“A phase-evolution based constitutive model that incorporates the dehydration-induced volume shrinkage and material stiffening was implemented into an ABAQUS (Dassault Systems, Johnston, RI, USA) user material subroutine (UMAT) (Supplementary Note 10) to simulate the mechanical behavior of the AP hydrogel²². Material parameters in the UMAT was determined by fitting the experimental results (Fig. 2f) that depict the effects of water content on dehydration strain and Young's modulus. In simulation, water content C was mimicked by temperature and coupled temperature-displacement analysis was performed with C3D8T elements in ABAQUS element library. Hydrogel and ceramic elastomer parts were assumed to be perfectly bonded. All the structural geometry and boundary conditions were kept to be consistent with actual experiments. Upon simulating the sintering of ceramic elastomer, element deletion technique was employed to remove the hydrogel layer.”

Minor points:

Comment 3.4: Lines 118: What is the mass fraction of PEGDA in the ceramic slurry, and how it is determined in the later experiments?

Response: We thank the reviewer for raising this question. The PEGDA is used as crosslinker in this work. The PEGDA content in the ceramic slurry has a great influence on the mechanical properties of the printed ceramic elastomer, especially the modulus and elongation at break. Here, the PEGDA content refers to the mass fraction of PEGDA in the BA-PEGDA resin mixture. The experimental result in **Figure R12 (Fig. 2c in the manuscript and Supplementary Figure 5)** shows that the ceramic elastomer with PEGDA content of 10 wt.% has a fracture strain of ~40%. Whereas, the FE simulation result in **Figure R10 (Supplementary Figure 8)** shows that the maximum tensile longitudinal strain (ε_n) of the ceramic elastomer layer is less than 10% when the hydrogel-ceramic laminate bends into an arc shape that approaches a full circle. Considering that the ceramic slurry with more PEGDA has higher printing accuracy, we choose a ceramic slurry with PEGDA content of 10 wt.% for the subsequent 4D printing experiments to obtain a better printing accuracy. Since the BA-PEGDA resin has a mass fraction of 20% in the ceramic slurry, the actual mass fraction of PEGDA relative to the entire ceramic slurry is 2%. We have specified the PEGDA content in the

ceramic slurry used for the direct 4D printing of ceramics in the revised manuscript. The modifications are highlighted in yellow.

“The ceramic elastomer slurry is composed of benzyl acrylate (BA), PEGDA and zirconia (ZrO₂) nano powders. The mass ratio of BA-PEGDA resin to ZrO₂ ceramic powders is 1: 4. The PEGDA is used as crosslinker, and its content relative to BA-PEGDA resin is adjustable in the range of 1-10 wt.%. We use the ceramic slurry with PEGDA content of 10 wt.% for direct 4D printing of ceramics.”

Figure R12. Mechanical properties of ceramic elastomer. a, Stress-strain curves of ceramic elastomer with different PEGDA contents (**Fig. 2c in the manuscript**). **b,** Young's modulus and fracture strain of ceramic elastomer with different PEGDA contents (**Supplementary Figure 5**).

Comment 3.5: Lines 119: Does the volume of ceramic elastomer layer change during dehydration and debinding?

Response: We thank the reviewer for raising this question. **Figure R13 (Supplementary Figure 4)** shows the photographs of hydrogel-ceramic laminate at different stages. In the dehydration process, the ceramic elastomer layer occurs bending deformation, and its volume can be approximated as unchanged. During the debinding process, the organics in the ceramic elastomer are decomposed along with the dehydrated hydrogel, and ceramic remains. In this process, the spacing between ceramic particles is slightly shortened. According to our experiments, after debinding at 800 °C, the entire ceramic structure has a linear shrinkage of ~3%, corresponding to a volume shrinkage of ~8.7%. After sintering at 1450 °C, the ceramic becomes very dense, with a linear shrinkage of 21.74%, corresponding to a volume shrinkage of ~52.1%. We have added the discussion about volume change of ceramic layer in the Supplementary Information (Supplementary Note 4 and Supplementary Figure 4).

Figure R13 (Supplementary Figure 4). Photographs of hydrogel-ceramic laminate at different stages. a, After dehydration. b, After debinding. c, After sintering. The insets show the SEM images. Scale bars, 5 mm.

Comment 3.6: Typos: Lines 129 and 501: “Ar2” to “Ar”

Response: We thank the reviewer for pointing out this typo. We have corrected this in the revised manuscript.

REVIEWERS' COMMENTS

Reviewer #1 (Remarks to the Author):

The authors have adequately addressed my earlier comments, so the manuscript could be accepted in its current form.

Reviewer #2 (Remarks to the Author):

No - the reviewer has no further comments to address and is satisfied

Reviewer #3 (Remarks to the Author):

The authors have satisfactorily addressed my concerns, and the revised manuscript is of better quality than the original submission. Therefore, the manuscript can be recommended for publication.

RESPONSE TO REVIEWERS' COMMENTS

Response: The authors would like to thank the reviewers for their valuable comments. Our point-by-point responses to each of the comments are as follows.

Reviewer #1:

The authors have adequately addressed my earlier comments, so the manuscript could be accepted in its current form.

Response: We thank the reviewer for those valuable suggestions and this positive comment on our revisions.

Reviewer #2:

The reviewer has no further comments to address and is satisfied.

Response: We thank the reviewer for those valuable suggestions and this positive comment on our revisions.

Reviewer #3:

The authors have satisfactorily addressed my concerns, and the revised manuscript is of better quality than the original submission. Therefore, the manuscript can be recommended for publication.

Response: We thank the reviewer for those valuable suggestions and this positive comment on our revisions.